# Assessing uncertainties in global cropland futures using a conditional probabilistic modelling framework

Kerstin Engström[1], Stefan Olin[1], Mark D.A. Rounsevell[2], Sara Brogaard[3], Detlef P. van Vuuren[4,5], Peter Alexander[2], Dave Murray-Rust[6], Almut Arneth[7]

[1] Department of Physical Geography and Ecosystem Science, Lund University, Sölvegatan 12, 22362 Lund, Sweden.
[2] School of GeoSciences, University of Edinburgh, Geography Building, Drummond Street, EH89XP Edinburgh, UK.
[3] Centre for Sustainability Studies, Lund University (LUCSUS), Biskopsgatan 5, 22362 Lund, Sweden.
[4] PBL Netherlands Environmental Assessment Agency, Postbus 303, 3720 AH Bilthoven, The Netherlands.
[5] Copernicus Institute for Sustainable Development, Faculty of Geosciences, Utrecht University, Heidelberglaan 2, 3584 CS Utrecht, The Netherlands.
[6] School of Informatics, University of Edinburgh Appleton Tower, 11 Crichton Street, Edinburgh EH8 9LE, UK.
[7] Karlsruhe Institute of Technology, Institute of Meteorology and Climate Research, Atmospheric Environmental Research (IMK-IFU), Kreuzeckbahnstr. 19, 82467 Garmisch-Partenkirchen, Germany.

*Correspondence to*: Kerstin Engström (kerstin.engstrom@nateko.lu.se)

**Abstract.** We present a modelling framework to simulate probabilistic futures of global cropland areas that are conditional on the SSP scenarios. Simulations are based on the Parsimonious Land Use Model (PLUM) linked with the global dynamic vegetation model LPJ-GUESS using socio-economic data from the SSPs (Shared Socio-economic Pathways) and climate data from the RCPs (Representative Concentration Pathways). The simulated range of global cropland is 893-2380 Mha in 2100 (± one standard deviation), with the main uncertainties arising from differences in the socio-economic conditions prescribed by the SSP scenarios, and the assumptions that underpin the translation of qualitative SSP storylines into quantitative model input parameters. Uncertainties in the assumptions for population growth, technological change and cropland degradation were found to be the most important for global cropland, while uncertainty in food consumption had less influence on the results. The uncertainties arising from climate variability and the differences between climate change scenarios do not strongly affect the range of global cropland futures. Some overlap occurred across all of the conditional probabilistic futures, except for those based on SSP3. We conclude that completely different socio-economic and climate change futures, although sharing low to medium population development, can result in very similar cropland areas at the aggregated global scale.

## 1 Introduction

Land use and land cover change (LULCC) is a fundamental aspect of global environmental change, but large uncertainties exist in estimating the effect of multiple drivers on LULCC in the future (Brown et al., 2014). A range of different models and scenarios have been used to project future cropland areas to 2100 with estimates in the range of 930 to 2670 Mha (Alexander et al., under review; Prestele et al., 2016). This compares with today's cropland areas of around 1530 Mha. The

large differences in these projections reflect uncertainties in process understanding, the use of different models to represent these processes and the direction of development of multiple drivers, including food demand and agricultural productivity (Schmitz et al., 2014; Smith et al., 2010). The direction of socio-economic drivers is referred to as deep uncertainties, which are addressed through the use of scenarios (van Vuuren et al., 2008). Cropland projections at the high-end of the projected

uncertainty range for global cropland would have profound consequences for, for example, global carbon and nitrogen fluxes, the global water balance, biodiversity and other ecosystem services (Lindeskog et al., 2013; Pereira et al., 2012; Zaehle et al., 2007). Hence, quantifying and understanding the inherent uncertainties in the drivers of LULCC has important consequences for policy responses to support sustainable development. However, the effects of uncertainties in the underlying scenario assumptions have not been systematically quantified for global cropland projections.

Scenarios are characterised by storylines that describe assumptions about key drivers and processes from which model input parameters are interpreted (Rounsevell and Metzger, 2010). These parameter interpretations are by definition deterministic within a scenario context, since they do not consider the uncertainties associated with the interpretation process itself. By contrast, probabilistic approaches examine system uncertainties by assigning probability distributions to input variables (reflecting uncertainties about scenario assumptions) to assess the influence of uncertainty on system outputs (van Vuuren et

al., 2008). The conditional probabilistic approach combines the strength of scenarios in addressing deep uncertainties with the probabilistic approach that explores the uncertainties in the assumptions about model input parameters (O'Neill, 2005; van Vuuren et al., 2008). In this case, the probability distribution of each model input parameter is conditional on the internal logic and assumptions within the contextualising scenario. Hence, conditional probabilistic futures are useful in exploring parameter uncertainty within and across scenarios (Brown et al., 2014; van Vuuren et al., 2008).

In this paper, we present probabilistic futures of global cropland that are conditional on scenario assumptions. In doing so, we quantify the uncertainties within these assumptions, as well as representing the deep uncertainty across different scenarios. The assessment is based on the following key questions: How will cropland area evolve until 2100 in response to socio-economic drivers and climate change? Will future ranges of global cropland for different scenarios overlap due to differences in socio-economic conditions and/or due to uncertainties in model input parameters? How does the influence of

the uncertainties in model input parameters change through time?

We use a scenario framework based on the five Shared Socio-economic Pathways (SSPs) and develop a scenario-matrix combining the five SSPs and four Representative Concentration Pathways (RCPs). This scenario-matrix is filled with probabilities based on the assumption that a given SSP will correspond with a given RCP. For each SSP, we derive RCP-specific input (yields in this case) applying the scenario-matrix. The resulting conditional probabilistic futures are named F1-

F5, where the numbers 1-5 correspond to SSP1-5. The RCP/SSP scenario framework was used since it is the most recent scenario approach for global environmental change research (Ebi et al., 2014; O'Neill et al., in press; van Vuuren et al., 2011; van Vuuren et al., 2014). We apply these scenarios to a global scale, socio-economic model of agricultural land use change (the Parsimonious Land Use Model; PLUM; Engström et al., 2016) in combination with crop yield time-series derived from the dynamic global vegetation model, LPJ-GUESS (Lindeskog et al., 2013; Smith, 2001). PLUM has been

benchmarked against different models and scenario studies in a land use model inter-comparison exercise (Alexander et al., under review; Prestele et al., 2016) that has demonstrated its consistency in comparison with other global cropland simulations. Because of its rapid runtimes, PLUM can explore uncertainties across its input parameter space (Engström et al., 2016), and hence is appropriate for use in probabilistic simulations requiring multiple model iterations.

**2 Materials and Methods**

**2.1 Conditional probabilistic futures within the SSP-RCP scenario framework**

To construct the conditional probabilistic futures (F1-F5) we used qualitative and quantitative information from the SSPs directly and quantitative information from the RCPs indirectly as input to PLUM (Fig. 1).

The SSPs describe plausible, alternative societal development pathways over the 21st century in the absence of climate
change or climate policies (O'Neill et al., in press; O'Neill et al., 2013). The SSP specific development of society and sectors, such as energy and land use, result in varying challenges for mitigation and adaptation to climate change (O'Neill et al., in press). In SSP1, economic growth and technological development are strong, sustainable solutions are preferred and population growth is low, resulting in low challenges for mitigation and adaptation. By contrast, SSP3 has high challenges for mitigation and adaptation because it is characterised by high population growth, but low economic and technological
growth, combined with resource intensive lifestyles. SSP2 describes a world with medium population growth, technological and economic development, resulting in medium challenges for mitigation and adaptation. The remaining two scenarios (SSP5 and SSP4) have contrasting challenges for mitigation and adaptation. SSP5 is a fossil fuel based world that is focused on development (low population growth, high economic and technological growth), has a high challenge for mitigation, but a low challenge for adaptation. SSP4 has a low challenge for adaptation, but a high challenge for mitigation because inequality
is high across and within countries, and various levels of economic and technological development benefit the global elite.

In the modelling framework presented here, the different socio-economic pathways were combined with different levels of climate change associated with the four RCPs. The RCPs are defined by their forcing targets from 2.6 to 8.5 W m$^{-2}$ at the end of the 21st century and trajectories of emission changes (van Vuuren et al., 2011). The radiative forcing of the RCPs are likely to correspond to global mean temperature increases between 0.3º C and 4.8º C by the end of the 21st century (2081-
2100) relative to the 1986-2005 global mean temperature (in the 5 to 95% range, Collins et al., 2013). The impact of climate change on crop yields was estimated for all RCPs by running LPJ-GUESS with climate inputs derived from five different General Circulation Models (GCMs: Collins et al., 2011; Dufresne et al., 2013; Dunne et al., 2013; Iversen et al., 2013; Watanabe et al., 2011).

The SSPs and RCPs were combined within a scenario matrix to reflect assumptions about the plausibility of an RCP arising
from a SSP. Theoretically, all cells within this matrix are possible, but not all combinations of SSPs and RCPs are equally plausible and consistent (van Vuuren and Carter, 2014; van Vuuren et al., 2014). For instance, van Vuuren et al., (2012) indicate that emissions are only likely to be as high as assumed under RCP8.5 with large-scale use of fossil fuels, driven

either by rapid economic growth (and little substitution towards less carbon-intensive fuels) or very high population growth. By contrast, for the matrix cells at the lower end of the RCP range (2.6 and 4.5 W m$^{-2}$), the SSPs would need to assume strong mitigation efforts, but this would be less plausible for the SSPs with high challenges to mitigation. In the SSP-RCP framework, climate mitigation policies can be prescribed by what are termed shared policy assumptions (SPAs). The SPAs define key attributes of climate policy, e.g. climate goals, policy regimes and measures (Kriegler et al., 2014). However, some SSP-RCP combinations remain unlikely either at the low or high end. Without the introduction of specific mitigation policies, the SSP-RCP combinations are referred to as reference scenarios (O'Neill et al., in press).

The conditional probabilistic approach was implemented through the following steps (van Vuuren et al., 2008):

1. Identification of uncertain parameters;
2. Estimation of the conditional probability ranges associated with these parameters, i.e. their Probability Density Functions (PDFs);
3. Use of Monte-Carlo sampling across the PDFs to undertake multiple simulations;
4. Identification of the uncertainty ranges in model outcomes and the determinants of model uncertainty.

Thus, the uncertainty ranges of the global model output variables arise from:

a. Uncertainties in the PLUM socio-economic input parameters; and,
b. Uncertainties in climatic variation and sampling from the scenario matrix for the crop yield simulations.

The effect of (a.) was explored in combination with (b.) for global output variables. We hypothesise that the effect of (b.) is smaller than the effect of (a.) on the range of global cropland change. This hypothesis was tested by undertaking model runs in which only the socio-economic parameter uncertainties were explored. This stepwise approach is described in more detail below (Sect. 2.3: steps 1-3 for a., Sect. 2.4: steps 1-3 for b, Sect. 2.5: step 4).

## 2.2 PLUM simulations – descriptions of future cropland

PLUM simulates agricultural land use in terms of cropland for 160[1] countries (Engström et al., 2016). The model is based on a simple demand and supply strategy, where demand of agricultural products is driven by population, economic development and dietary changes. The supply of agricultural products, indicated by cereals, is met by a simple trade mechanism, where all countries are assumed to have access to the global market. Changes in cropland are assumed to be proportional to changes in cereal land, applying a constant country-specific cropland-cereal land ratio derived from FAOSTAT data in the baseline year of 2000 (for a detailed description of PLUM see Engström et al., 2016). Globally, cereal land alone accounted for 60% of cropland in 2010 (FAOSTAT, 2015). PLUM can reproduce global historic agricultural land use change (1990-2010), and results have demonstrated that agricultural land use is highly sensitive to uncertainties in crop yield growth rates (Engström et al., 2016). To estimate the temporal trends and changes in spatial patterns of crop yields in response to climate change, country level cereal yields were derived from the global dynamic vegetation model LPJ-GUESS (Lindeskog et al., 2013;

---

[1] The availability of input data determines the number of countries included. In the evaluation version in Engström et al. (2016) 162 countries were included, whereas 160 are used here.

Smith, 2001). LPJ-GUESS accounts for the effects of temperature, precipitation and atmospheric $CO_2$-concentrations on crop yields and the productivity of natural vegetation. It models the yields of 11 globally important crops, including wheat, maize and rice (Lindeskog et al., 2013) and also accounts for management options such as sowing and harvesting in response to climatic conditions.

To derive LPJ-GUESS country level projections of actual and potential cereal yield, LPJ-GUESS simulations were performed on a 0.5 degree grid from 2000 until 2100 for four cereal crops (wheat, maize, millet/sorghum and rice), both rain fed and irrigated. To calculate the yearly actual yield per grid cell, the per grid cell share of irrigated vs. rain fed area in the year 2000 was derived from the MIRCA data set (Portmann et al., 2010) and applied over the entire simulation period. For potential yields, we chose the maximum of either rain fed or irrigated yields in each grid cell where the crop was present in

the MIRCA data set. Naturally in most cases irrigated yields are larger and less fluctuating. The per grid cell actual and potential yields for the year 2000 were then scaled to the grid cell actual and potential yield from Mueller et al. (2012) in order to establish the difference (yield gap) between actual and potential yields. This scaling factor was then used for all years in the yield time series. The yield time series were then aggregated to the country level based on the area fractions from the MIRCA data set. These aggregated county level time-series of actual and potential yield were used to sample yield time-

series as input to PLUM (Sect. 2.4.3). In PLUM, the yield gap was modelled to change over time depending on three scenario parameters describing technological change (Fig. 2).

These three parameters describe the strength of technological change per se and the investment and distribution of yield improving management practices (see Appendix A). The country level actual yields calculated in the previous step might differ slightly from the country statistics from FAOSTAT on which PLUM is based. Thus, as a final step, the yield

calculated in PLUM was scaled to match yields reported in the year 2000 (FAOSTAT, 2015). The scaling factor for the year 2000 was applied throughout the simulation period. An example of the yield calculations is given in Fig. 2, for Ukraine. This example uses the socio-economic assumptions from SSP5 and the yield projections driven by RCP6.0. Carbon fertilization has a strong effect on crop yields, e.g., for Ukraine potential yields are simulated to increase from below 6 t ha$^{-1}$ to above 8 t ha$^{-1}$ from 2000 to 2100 (dark grey dashed line, Fig. 2). Actual yield is simulated to increase by roughly 1 t ha$^{-1}$ by the end of

the 21$^{st}$ century (light grey dashed line, Fig. 2). However, the strong economic and technological change in SSP5 results in a tripling of yields during the period 2000-2100 (grey line, Fig. 2) and thus a decrease in the yield gap for Ukraine.

## 2.3 Uncertainties related to the socio-economic input parameters of PLUM

### 2.3.1 Identifying uncertain parameters in PLUM

The computational costs of PLUM are relatively low since it is a simple model that operates at the country scale, with global parameterisation and a focus on aggregated global outputs. This allows a wide range of socio-economic input parameters to

be tested. Thus, in addition to input parameters that affect cropland changes directly, we also analysed input parameters that affect other global output variables such as meat consumption and cereal demand.

### 2.3.2 Assessment of the conditional probability ranges informed by the SSPs

The conditional probability ranges describe the uncertainty (± 1 standard deviation) around the mean of each input parameter. The following describes the assessment of the conditional probability ranges both for input parameters parameterised with country level time-series (Nr. 0, see Table 1: economic development indicated by Gross Domestic Product "*gdp*" and population "*pop*") and global socio-economic input parameters (Nr. 1-12, see Table 1 and Table 2).

For each SSP, the mean for the parameters *gdp* and *pop* was specified using the country level projections from the SSP-Database (SSP-Database, 2015). We used the population projections "IIASA-WiC v9_130115" from 2010 to 2100 (KC and Lutz, in press; SSP-Database, 2015) and combined these with population data from the World Bank for the years 2000-2009 (World Bank, 2015). For the economic development projections, the "OECD Env-Growth v9_130325" from 2000-2100 were selected (SSP Database, 2015) that have the advantage of providing the country- specific PPP (Purchasing Power Parity) -MER (Market Exchange Rate) conversion rates required by PLUM.

The applied population and GDP projections are SSP and country specific, but retain uncertainty with respect to interpretation of the underlying drivers, model structures and country groupings. These uncertainties were explored with the uncertainty levels of the global parameters *gdp* and *pop* (Table 1). The uncertainty levels of *gdp* were orientated on the coefficients of variation calculated from three different projections for global GDP available in the SSP database (SSP-Database, 2015) and set to be 7%, 7.5%, 3.5%, 12.5% and 7% for SSP1 to SSP5, respectively. For population projections, differences in input assumptions and models resulted in coefficients of variation between 2-21% in 2100 (SSP-Database, 2013; SSP-Database, 2015; O'Neill, 2005). Population projections are very sensitive to fertility rates (Lutz and KC, 2010), so qualitative uncertainty levels (low, medium, high) were estimated based on the heterogeneity of assumptions for different fertility groupings (high fertility countries, low fertility countries and rich OECD countries; KC and Lutz, in press; O'Neill et al., in press). The low, medium and high uncertainty levels were set to be ±2%, ±4% and ±6% of total population size in 2100, respectively, see Table 1.

For the mean values of the other global model input parameters we started with the historic mean value for each parameter (Engström et al., 2016) and assessed a baseline trend qualitatively (Table 1). The positive or negative strength of the qualitative baseline trends were characterised with symbols (---, --, -, 0, +, ++, +++). Similarly, the changes in trends were estimated for each SSP based on an interpretation of the SSP storylines. For transparency, we provide a summary of our interpretation of the SSPs based on the existing SSP narratives (O'Neill et al., in press) (see Appendix B). We also recorded the scenario elements of the existing SSP narratives (O'Neill et al., in press) that were assumed to influence changes in the PLUM input parameters (Table 2).

Low, medium or high uncertainty levels were attributed to each input parameter and scenario. These uncertainty levels comprise several sources of uncertainty: the understanding of the world characterised by a storyline, the knowledge about the

global average development of a driver, and the heterogeneity and variability of the model parameter across and/or within countries. The change in trend and uncertainty level (see Table 1) were interpreted for each model input parameter conditional on each SSP using the scenario elements in Table 2. For example, we assumed that the scenario elements of "Technology development & Transfer" and "Agriculture" (Table 2) influence the input parameters of yield development (for

both cereals and animal products: *fcr improvement*, *technology, investment* in Table 1).

For SSP5, technology development and transfer are described as being rapid and agriculture is highly managed and resource-intensive with a rapid increase in productivity (O'Neill et al., in press). We interpreted this as strong improvements in feed conversion ratios (5: *fcr improvement*: +++, Table 1) and a strong trend in investments and technology for yields (6: *technology*: +++ and *investments*: +++, Table 1).

Some PLUM input parameters are not global, but based on country groups to reflect variability in local contexts, such as meat consumption (Table 1, Nr. 3: meat1-4). For example, in SSP1, the scenario element "consumption and diet" is described as "low growth in material consumption, low-meat diets, first in HICs (High Income Countries)" (O'Neill et al., in press), resulting in differentiated estimates for countries grouped in meat1 (traditionally high meat consumption) compared to countries belonging to meat3 and meat4 (transitioning and low income countries (see Engström et al. (2016) for details on

consumption classes). More detailed examples of the logic in assigning changes in trends and uncertainty levels to input parameters conditional on each SSP are provided in Appendix B.

These qualitative estimates of changes in trend and uncertainty levels for the PLUM input parameters in Table 1 were translated into quantitative values (mean and standard deviation characterising the PDF, see Sect. 2.3.3) by sampling from an input parameter value matrix (input parameters in rows, symbols --- to +++ in columns, see Appendix B, Table B1).

**2.3.3 Monte-Carlo sampling of socio-economic parameters**

To assess the uncertainty in projected model output, Monte-Carlo sampling was used to create different sets of PLUM input parameters from PDFs conditional on each SSP. We assumed a normal distribution for most PLUM input parameters since it seems unlikely that extreme values would occur frequently. Moreover, extreme values would be applied to all countries simultaneously due to the nature of the global parameterisation (i.e. in one model run all countries have the same value). The

choice of normal distributions was also supported by the normal distribution seen in the inter-country variability in historic data for global parameters of e.g. meat and milk consumption. The land conversion parameters (Nr. 7-9, Table 1) are an exception, as their values only limit the internally calculated land conversion rates. Each maximum value was thought to be equally plausible and so we assumed the land conversion rates to be uniformly distributed. The PDFs were constructed by using the mean and standard deviation (min and max for land conversion parameters) derived for each SSP (Table B1,

Appendix B). All PDFs were truncated by minus three standard deviations at the lower end and plus three standard deviations at the higher end of the distributions. In some cases, the PDFs were truncated by one standard deviation, e.g. for the lower bound of population in SSP1, as it was assumed unlikely that population would decrease much more than projected for this scenario (see Appendix B). For each iteration, a random number was drawn to calculate an input parameter from the

appropriate PDF. All countries were assumed to draw the same value from the same distribution of parameter values for each run. This simplified approach could lead, arguably, to an overestimation of uncertainties, since in reality between-country differences in deviations from the mean would be expected. In the uncertainty analysis we did not investigate the possible effects of correlation between input parameters because the use of scenarios ensures the consistency across parameter mean

values. For example, in SSP1, the relatively high mean values for all three input parameters that influence yield development (*distribution*, *technology*, *investment*, see Table 1) are all consistent with the storyline of relatively strong technological growth and technology transfer. Two sets of Monte-Carlo simulations were performed. For the first set only the socio-economic parameters were sampled and the mean yield of each SSP (derived from the RCP-SSP matrix, see Sect. 2.4.3) was used (3600 runs per SSP). For the second set, in addition to the socio-economic parameters, crop yields were also sampled

based on the combined uncertainty from the RCP-SSP matrix and GCM variability, see Sect. 2.4.3. Because of the increased sampling in these simulations the number of iterations was increased to 7200 per SSP.

## 2.4 Uncertainties arising from climate change and climate variability

### 2.4.1 Identifying parameters informed by the RCPs

The RCPs cover a wide range of emission and concentration scenarios; at the low end with the mitigation pathway RCP2.6

and at the upper end with the high emission pathway RCP8.5 (van Vuuren et al., 2011). For a given RCP, modelled global average temperatures between different GCMs can vary by up to 1ºC in 2100. The global totals and spatial patterns of other climatic variables, e.g. precipitation, also vary strongly between GCMs (Amiro et al., 1999; Knutti and Sedlacek, 2013). The effect on the global terrestrial carbon balance of between-GCM variability can be larger than the difference between concentration pathways (Ahlström et al., 2013). Thus, a potentially important source of uncertainty in the crop yield

projections is the climate variability projected by the different GCMs.

A second source of uncertainty in future crop yield projections is that each SSP could, though with different probabilities, lead to different RCPs. To address this uncertainty, the likelihood of SSP-RCP combinations was estimated (in the absence of mitigation strategies) as described below.

### 2.4.2 Assessment of the SSP-RCP scenario-matrix

The SSP-RCP probability judgements were combined with the interpretation of the SSP storylines (O'Neill et al., in press) to estimate the conditional probabilities (van Vuuren and Carter, 2014) given in Table 3. The sustainability assumptions in the SSP1 scenario with respect to environmental and energy policies could curb emissions sufficiently to achieve RCP2.6, but it is more plausible for the SSP1 scenario to arrive at greenhouse gas concentrations consistent with RCP4.5 and RCP6.0 (medium probability).

Many medium reference scenarios result in forcing levels around 6-7 W m$^{-2}$ based on a continued reliance on fossil fuels and medium population and economic growth (Clarke et al., 2014). We interpreted this as a high likelihood of forcing levels

similar to RCP6.0 being achieved by SSP2, SSP3 and SSP4. High energy intensity in low income countries and material-intensive consumption make RCP8.5 plausible in a SSP2 world, although this is assumed to have a low probability of occurrence. Given the relatively low economic growth in SSP3, we assume that forcing levels would lead to RCP6.0 or RCP8.5, with a lower probability for RCP8.5 (Table 3).

The very high emissions pathway of RCP8.5 can only be achieved with a combination of, for example, high economic growth and reliance on fossil fuels. The divergent development in SSP4 for the few elite and the many fewer privileged people is difficult to estimate. We assumed that SSP4 has a high probability of resulting in forcing around RCP6.0 or possibly lower. The latter is based on the moderate population growth and the original positioning of the scenario (low mitigation challenge). The majority of the population in SSP4 cannot afford a material-intensive lifestyle, making RCP8.5

forcing unlikely. For SSP5 we assumed that the material-intensive lifestyle combined with very high economic growth would lead to RCP8.5 with a high probability (comparable to the assumptions for the SRES A1F1 scenario, see van Vuuren and Carter, 2014).

The qualitative probabilities in Table 3 were translated into quantitative values in Table 4. We assumed that the qualitative notions of very high, high, medium, low, and very low probability translated into quantitative probabilities of 0.9, 0.75, 0.5,

0.25 and 0.1 respectively. The assigned probabilities were normalised so that the sum of probabilities for each SSP equalled 1 (see Table 4).

### 2.4.3 Sampling the climate driven parameters: yields

The probabilities of SSPs resulting in RCPs (Table 4) were combined with the uncertainties arising from the climate variability of the different GCMs. To do so, the aggregated country level yield time-series (described in Sect. 2.2) were

calculated for each RCP-GCM combination. Yield time series calculated for different countries vary, depending on the underlying GCM. To account for this spatial variability, the deviations from the yields averaged per RCP ($\hat{Y}$) were decomposed using singular value decomposition (SVD, Eq. 1), where $Y$ is the yield projections for each GCM-RCP combination, $\hat{Y}$ is the mean over the GCM projections for each RCP and $U$, $S$ and $V$ are factors that can be used to reconstruct $Y - \hat{Y}$.

$$U, S, V = SVD(Y - \hat{Y}) \tag{1}$$

This allows sampling of per country yield projections while preserving the patterns in spatial variability resulting from the GCM-RCP yield projections. From this we constructed four sets (one for each RCP) of 51 future yield projections, where 50 are random samples calculated using (Eq. 2) where $V$ is replaced with $\varepsilon \in N(0,1)$ and one set with the mean yield across the GCMs ($\varepsilon = 0$). 51 samples were chosen to allow enough variability in the effect of the GCMs on the yield projections.

$$Y = \hat{Y} + US\varepsilon, \quad \varepsilon \in N(0,1) \tag{2}$$

Drawing from these four sets, the SSP-RCP matrix was used to weigh how much of the information from the different RCPs should be taken into account for each SSP. The resulting yield time-series ($y\_i$) were sampled using a uniform distribution ($y\_i$, $i \in U(0,50)$) and the sample used as input to PLUM together with the socio-economic parameters (see Sect. 2.3.3).

## 2.5 Analysis of uncertainty results and sensitivity assessment

Cropland distributions with and without climate variability were analysed for the 7200 and 3600 runs respectively to the year 2100 with respect to convergence and or divergence across the five scenarios. We report the mean development and ranges at 95% confidence levels (corresponding to two standard deviations) for the global-scale, model outputs, i.e., population, GDP, cereal consumption, meat consumption, feed demand, cereal demand, cereal production, cereal yield and cropland. The word "likely", based on the IPCC's recommendation for uncertainty communication (Mastrandrea et al., 2010), was
used to report results that are probable at 68-100%. A global sensitivity analysis (GSA; Saltelli et al., 2008) was carried out to quantify the contribution of the input parameters to the uncertainty of the global cropland extend for all socio-economic model input parameters. The GSA was implemented as previously described in Engström et al. (2016), with n=5000, p=24, requiring 130 000 runs for each scenario (according to (p+2)*n= number of runs; Lilburne and Tarantola, 2009). We used the soboljansen method (Pujol et al., 2014), which is an R implementation of the Monte Carlo Estimation of Sobol sensitivity
indices (Jansen, 1999; Saltelli et al., 2010),  as a method that is robust for large and small total indices (Saltelli et al., 2010). The total indices (hereafter, total importance) consist of the first-order effect of each input parameter and their interaction effects. We excluded the parameters that describe the allocation of cropland changes to forested or grassland areas (*grassForest*) and the forest degradation rate (*forestDeg*) for the global sensitivity analysis because they have no direct impact on global cropland. We used the GSA to visualize the total importance, which describes the main effect and
interaction effects of the uncertainty for each input parameter on the model output (cropland).

## 3 Results

### 3.1 Ranges of global cropland projections

Global cropland area increases initially for all scenarios, but declines in the simulations for F1 after 2015 (Fig. 3, Panel a). F1 continues to decline to 963 ± 140 Mha of global cropland by 2100. All other cropland futures increase until 2030;
thereafter the rate of increase slows for F2, F4 and F5. Mean global cropland peaks for F5 in 2045, and shortly afterwards for F2, and then decreases to 1400 ± 382 Mha in 2100 for F5 and 1590 ± 332 in 2100 for F2. For F3, global cropland continues to increase over the entire simulation period reaching 2280 ± 200 Mha in 2100. Mean global cropland changes for F4 are very moderate throughout the simulation period and are within the cropland development of the other scenarios (1540 Mha in 2100). However, the range of cropland futures for F4 for the 95% confidence interval in 2100 is very wide (1126-1954
Mha). The cropland distribution of F4 overlaps with the cropland distributions of all other scenarios, as do the cropland

distributions of F2 and F5. F1 by contrast has the smallest cropland range, which is also indicated by its peaked distribution (Fig. 3, Panel b).

The cropland distribution for F1 is skewed toward the higher end, which is due to the truncated distribution of uncertainties in the population projections. For the same reason, the distribution of F3 is slightly skewed toward the lower end. The cropland distribution in F3 is also peaked, indicating that the confidence in the model outcomes for F1 and F3 are the highest, despite the fact that these two scenarios show divergent global cropland development.

The variability in yields arising from the five GCMs and sampling from the SSP-RCP matrix does little to change the shape of the global cropland PDFs (Fig. 3, Panel b, comparing solid lines (with yield variation) to dashed lines (mean yield)). For cropland futures with flat distributions (F2, F4 and F5), the distributions with climate variability (solid lines) are slightly less peaky than without climate variability (dashed lines). This indicates that the climate variability contributes more to the overall uncertainty of global cropland areas for scenarios with larger overlap of global cropland outcomes (F2, F4 and F5), compared to the cropland futures F1 and F3. Overall, the effect of climate change variability and sampling from the SSP-RCP matrix is very small. However, the inter-annual variability of yields due to variations in climate patterns is considerable (not displayed here).

## 3.2 Socio-economic dynamics influencing cropland futures

The strongly overlapping cropland ranges for the F2, F4 and also F5 scenarios are caused by the assumed uncertainties in the trends of the different input parameters, but also by the counteracting effect of different scenario drivers leading to similar cropland futures. Conversely, the distinct development of cropland for the F1 and F3 scenarios is mainly due to the reinforcing dynamics of drivers as described below.

In contrast to the other population scenarios, the total population size in F3 does not peak in the 21$^{st}$ century, but grows continuously to 12.1 ± 1.5 billion people by 2100 (at the 95% confidence interval, Fig. 4, Panel a). This steady increase in population influences cereal consumption, cereal demand and (less clearly) cereal production (Fig. 4, Panel c, f, and h) and thus cropland (Fig. 3). The strong population growth and therefore high food demand is counteracted by the low economic growth in F3, which results in the relatively lower consumption of animal products (Fig. 4, Panel d), corresponding to 53 kg of meat per person in 2100. However, in spite of this, slow technological change reinforces the high demand for cereals due to the steady increase in cereal feed (Fig. 4, Panel e).

For F5, despite a decline in total population size to 7.4 ± 0.9 billion people by 2100, the consumption of animal products by 2100 is 820 ± 150 Mt meat and 1230 ± 242 Mt milk. The former corresponds to an average meat consumption of 110 kg meat per person in 2100, which is comparable to current meat consumption rates of several developed countries, e.g., the US, Australia and Austria (FAOSTAT, 2015). The consumption of animal products is driven by economic growth and a very resource-intensive lifestyle for all consumption groups. For scenarios with strong technological growth, i.e. F1 and F5, the efficiency of the production of meat and dairy products increases and thus the demand for total cereal feed decreases.

Likewise, strong economic growth and technological change result in high global average cereal yields in 2100 for F1 and F5, $5.4 \pm 0.5$ t ha$^{-1}$ and $5.6 \pm 1.0$ t ha$^{-1}$, respectively (Fig. 4, Panel g). For F5, the strong technological growth and resulting high yields and the high consumption levels balance the need for global cropland changes. For F1, the high yields and low consumption levels reinforce the diminishing need for cropland. By contrast, for F3, the increase in yield from 3.1 to $4.1 \pm 0.7$ t ha$^{-1}$ and the expansion of cropland from 1503 to $2280 \pm 200$ Mha in the period 2000-2100 is not sufficient to keep up with rising cereal demand (including the demand from overproduction). In 2100 global cereal demand for F3 is $4550 \pm 718$ Mt, but production is $3960 \pm 814$ Mt. This inadequate global production would lead to cereal shortages and in a few cases to countries approaching their maximum available arable land. More importantly, the underproduction is due to insufficient, though conceptually consistent with SSP3, incentives for exporting countries to increase their production, as well as cropland degradation.

### 3.3 Uncertainty in socio-economic model inputs

The uncertainty in input parameters contributes differently to the uncertainty of global cropland futures over time (Fig. 5). For F1, population projections and technological change dominate uncertainty, the latter being especially important during the first quarter of the simulation period. Similarly, for F2, uncertainties in technological change and consumption are at first important, but after 2025 cropland degradation contributes largely to the uncertainty of global cropland. By contrast, population projections and technological change are the major contributors to the uncertainty range of global cropland for F3. For F4, uncertainties in the extent of land degradation, but also population projections and consumption and technological change contribute to uncertainties in global cropland. Consumption and technological change become less important over the 21$^{st}$ century, compared with land degradation and population. These trends are similar for F5.

### 4 Discussion

For F2, F4 and F5, the uncertainty distributions of global cropland overlap greatly, with cropland changes over the 21$^{st}$ century within the range of $-20\%$ to $+17\%$. This large overlap can be explained by counteracting drivers, but also by larger uncertainties in the assumptions of model input parameters for the SSPs with contrasting directions of change in challenges for mitigation and adaptation (i.e. SSP4 and SSP5). By contrast, the F1 and F3 cropland futures are very distinct from one another with a higher level of confidence, indicated by their peaked distributions. However, the discrepancy between total demand and production in F3 indicates that a focus on regional production with limited trade can risk food insecurity for countries with limited potential for domestic production, which concurs with Brown et al. (2014).

In F1 and F3, the total simulated range of cropland areas increases considerably from $-41\%$ to 58% in 2100 compared with 2000. These results lead to a slightly larger uncertainty range compared with deterministic scenario projections. For example, the cropland changes simulated by four different Integrated Assessment Models (IAMs) were 1130-2100 Mha cropland by 2100 (RCP4.5-GCAM and RCP2.6-IMAGE; Hurtt et al., 2011). This corresponds to cropland changes from

1990-2100 of -25% to +39% (Hurtt et al., 2011). However, it is difficult to compare these results directly since the IAM scenarios also include land based mitigation options. For example, the cropland changes of RCP2.6-IMAGE were the result of a stringent mitigation scenario, where the production of biomass for bioenergy increased cropland areas (Hurtt et al., 2011).

No climate change mitigation actions were assumed in this study, although for SSP1 this would be plausible and consistent with the storyline. The simulated decrease in cropland for F1 suggests that land-based mitigation options, such as bioenergy production, could be implemented on abandoned cropland without compromising food security or the provision of other ecosystem services. However, the global sensitivity analysis showed that for F1 to consistently achieve strong decreases in cropland areas it is important to stay within the range of input assumptions. Among others, consumption patterns have to
reflect the more resource efficient and environmentally friendly lifestyle that underlies this scenario. Achieving technological change and thus yield increase is important, as is decreased environmental degradation and thus decreased cropland degradation rates.

The LPJ-GUESS yield projections are at the higher end of the range of yield projections compared with other models (Rosenzweig et al., 2013) and likely overestimates the effect of $CO_2$ fertilization since nitrogen limitations were not included
in earlier versions of the model. However, these effects were counteracted in PLUM by: a) dividing global production by global cropland area to derive global average yield, which does not account for double cropping, and b) assumptions about cropland degradation that are implemented as a production loss, which decreases the simulated global average yield. Future research will consider the management options in LPJ-GUESS coupled to PLUM (e.g. the use of irrigation and fertilization scenarios) and will improve the potential impacts of climate change on yields arising from pests and heat stress. Currently,
heat stress implementation in LPJ-GUESS is limited to a shortened growing season, increased respiration and lowered photosynthesis.

The sensitivity analysis showed that assumptions about cropland degradation were important for cropland development across all scenarios. Cropland degradation was assumed to lead to an average global production loss of between 6% (F1) and 14% (F5) in 2100. This compares with an estimated global average of 20-40% loss of potential production on degraded
agricultural areas only (Zika and Erb, 2009). Hence, the PLUM results of total global production (not only on degraded agricultural areas) appear to be of the right magnitude and the sensitivity analysis highlighted the importance of accounting for these uncertainties.

Global cropland was less sensitive to the uncertainties associated with the consumption input parameters, which, for example, describe the rate of increase or decrease in meat consumption for the four consumption country-groups. PLUM
represents cultural differences in consumption patterns between countries (based on four consumption groups), but this could potentially mask part of the total importance of the consumption input parameters because the correlation between the parameters of the four consumption groups was not considered. Additionally, cropland changes are likely to be underestimated in F5 because meat consumption increases strongly in countries currently defined as developing and global average meat consumption approaches 110 kg per person in 2100. This would probably be associated with the intensification

of animal production, which currently is not included in PLUM. Since intensive meat production would lead to an increase in the feed share derived from cereals, cropland areas would increase.

The use of a global model with reduced complexity risks missing potentially important dynamics and feedbacks, which could affect the magnitude of change (e.g. intensification in the livestock sector, as highlighted above). A reduced complexity model could also widen or limit the uncertainty range in outputs (depending on the balance between introduced uncertainty and better overall model performance). A further potential drawback of this approach is that the estimated uncertainties for global model input parameters are largely judgemental. For example, a challenge in assessing these uncertainties was the high degree of variability across 160 countries. This was especially the case for SSP4 due to the assumed inequality within and across nations that leads to a wide range of cropland futures. However, these assumptions are at least reported in a transparent way. Overall, the conditional probabilistic approach applied in PLUM led to cropland area ranges that are consistent with those reported by other scenarios and model inter-comparison studies (Alexander et al., under review; Hurtt et al., 2011; Prestele et al., 2016; Schmitz et al., 2014), which provides confidence in the modelling framework. PLUM is based on cereal demand and assumes that changes in cereal land are a reliable proxy for food demand and cropland changes with free trade contributing greatly to meeting demands. For example, a change in the demand of cereals compared to other crops driven by climate change (either directly, or by enhanced demand for bioenergy) will require a revision to the constant cereal-cropland ratio. Future model development will take bioenergy production into consideration. The global scale projections with PLUM need to be interpreted under the assumption that the future agricultural system will not be fundamentally different from how we understand it today; an assumption that occurs in most global models. Clearly, in some scenarios the free trade simplification might not be valid (e.g., in SSP3), a limitation that is balanced by PLUM having simple and transparent relationships. We argue that the possibility to perform rapid model runs outweigh drawbacks in the current model version that arise from less than perfect regional model performance.

High-end climate change impacts on yields (i.e. from solely applying RCP8.5) were not tested here, as the goal of this study was not to assess the impact of each emission pathway on cropland, but instead to create plausible and consistent cropland futures that address the uncertainties within each scenario. This, and the importance of the technological change parameters in closing the yield gap, explains why the variability in climate change was found to have a small impact on global cropland areas. Small differences in the climate change impacts on agricultural areas between RCP4.6 and RCP6.0 were found elsewhere (Wiebe et al., 2015), as well as the comparatively larger effects of RCP8.5. However, using the scenario matrix populated with probabilities streamlined the total number of scenarios and simultaneously removed the need to compromise with single selections of SSP-RCP combinations.

**5 Conclusion**

Considering the simple supply and demand mechanism in the model, the use of cereals as a proxy for demand and area changes, the likely range of global cropland simulated in this study ranged from 893 to 2380 Mha in 2100. This was consistent with the range reported in the literature of 930-2670 Mha in 2100, although slightly skewed to the lower end of this range. This shows that uncertainties in input assumptions are equally important for output ranges than difference in model structures and that the entire uncertainty of global cropland development is probably even larger, if the sources of uncertainties are combined. For the uncertainties in input assumptions, we found that the main uncertainties in global cropland projections are the differences in the socio-economic assumptions of the scenarios, and the uncertainties in interpreting model input parameters. Cropland futures where the output PDFs did not overlap with other scenarios were found for the SSP1 scenario projections and the SSP3 scenarios, whereas the SSP2, SSP4 and SSP5 scenarios were found to have large areas of overlap. This was partly due to the compensating dynamics of drivers, e.g. strong yield development and increase in consumption in the SSP5 scenario, but also due to the larger uncertainties in scenarios with contrasting challenges for mitigation and adaptation (i.e. SSP5 and SSP4 scenarios). Uncertainties in population projections, technological change and cropland degradation were found to be the most important for uncertainty in global cropland projections, while uncertainties in consumption levels and production levels were found to be less important. When taking account of the uncertainty ranges at the 95% confidence interval across all scenarios, there were fewer differences between the scenarios, i.e. there is overlap at some level of probability in all global cropland projections, except for projections based on SSP3. This leads us to conclude that very different worlds, although sharing low to medium population development, can result in very similar cropland futures at the aggregated global scale.

**Authors contribution**

KE, MDR, AA, SO and DMR designed the study. KE and SB assessed the uncertainties for global PLUM parameters. DvV, KE, MDR, and PA estimated the qualitative and quantitative probabilities of the scenario matrix. KE and SO developed the model code and performed the simulations. KE prepared the manuscript with contributions from all co-authors.

**Acknowledgements**

This study was carried out under the Formas Strong Research Environment grant to AA, Land use today and tomorrow (LUsTT; dnr: 211-2009-1682). The author would also like to thank J. Lindström for discussion of statistical methods for the yield sampling. MDR, DvV and AA acknowledge support by the FP7 project LUC4C (grant no. 603542).

**Appendices**

**Appendix A: Model development**

**A1 Changes compared to previous PLUM version**

In comparisons to the version described in (Engström et al., 2016) several minor alterations (Table A1) and one larger
5   alteration were made to the model. The larger alteration relates to the representation of the yield development in PLUM, which is explained below. Assumptions are also made within PLUM about the scenario dependency of the availability of potential arable land (*residualNV*), reflecting different environmental policies in the SSPs.

**Table A1: PLUM development, affected variable, rational for development and implemented development.**

| Variable | Rational for development | Development |
|---|---|---|
| Food conversion ratio (*fcr*) | The *fcr* for beef, pork, mouton and chicken were input parameters that based on assumptions related to technological development were changed. The input parameter fcr improvement (*fcrImp*) was changed simultaneously for the same reason. This was a doubling of the effect of technological change on the efficiency of animal production. | The *fcr* input parameters were removed (but kept in PLUM as initial values) and *fcrImp* is the only scenario variable that changes animal production efficiency. |
| Expected production (*expPro*) | In the variable *expPro* it is calculated how much more cereals should be supplied by each country in the next year. The amount of cereals to add/subtract from current production is either the domestic cereal deficit/surplus or the country's share on the global cereal deficit/surplus. However, in the previous version the change in demand other than through increase in modelled food consumption (that is the demand externally created with the parameter *overProdRate*) was misleadingly not included. This is corrected in this version. | The *expPro* includes now the following rules: Global cereal surplus, exporting countries are assumed to decrease their production by the minimum of either their domestic surplus or their share on the global surplus (including demand created by *overProdRate*). Global cereal surplus, importing countries, no change is assumed, only if cereal-self-sufficiency should be increased countries are assumed to increase production by their domestic deficit. Global cereal deficit, exporting countries are assumed to increase their production by their share on the global deficit (including demand created by *overProdRate*). Global cereal deficit, importing countries are assumed to increase their production by the maximum of either their domestic deficit, or the share on the global deficit (including demand created by *overProdRate*). |
| Expected production (*expPro*) and residual natural vegetation (*residualNV*) | Previously there was no restriction regarding how much more cereals a land can be expected to produce based on the availability of land with natural vegetation (*grassland* and *forest*) in the countries. This was implemented here, using estimates of per country potential arable land (FAO, 2000). | A scenario dependent share of land with natural vegetation (specified in the input parameter residual natural vegetation (*residualNV, %* of potential arable land *potArableLand* (1000 ha, derived from FAO, 2000)) is restricting the *expPro* for countries. If a country has less residual natural vegetation (defined as all ((*forest+grassland)/potArableLand)\*100* or if ((*forest+grassland)/potArableLand)\*100* < residual potential arable land (*resPotAL = ((potArableLand-cerealland-restCropland)/potArableLand)\*100*) then *resPotAL*) left than *resiudalNV*), than no *expPro* is assumed for this country. Instead this country's *expPro* is divided on all other exporting countries. |
| Cropland degradation (*croplandDeg*) | Previously it was assumed that degraded cropland would be removed from the used cropland (that is *cerealland*), but it seems closer to reality that *croplandDeg* should influence the production capacity of the used *cerealland*. | *croplandDeg* changes the production of cereals with the following equation: *cerealProduction= cerealland\*cerealYield-cerealland\* cerealYield\*croplandDeg*/100\*time() So the value of *croplandDeg*, that is the share of lost production, is achieved at the end of the simulation period (after 100 years). |
| Cropland (*cropland*) | Earlier only *cerealland* was included. Now cropland was estimated. | *Cropland* was estimated assuming that the share of cereals and other crops (oilcrops, pulses, roots and tubers, vegetables and fruits) will remain constant in the future. |
| Forest (*forest*) and grassland (*grassland*) | Not only changes in *cerealland*, but also changes in *cropland* should affect *forest* and *grassland*. This is included here. | One additional stock *restCropland* is added, with flows from *grassland* and *forest*. *restCropland* is *cropland-cerealland*. The flow from grassland and forest equal (*landconversion/shareOfCropland – landconversion*)\* |

**A2 Yield development in PLUM**

The global parameter *6_technology* describes the change in trend of technological development. The parameter *6_investment* characterises how much yield increases as a function of GDP per capita. The parameter *6_distribution* describes how agricultural management practices are assumed to be transferred across and within countries. For example, in a scenario with an emphasis on human development it is assumed that the distribution of technologies would be more efficient and thus the yield gap would decrease more rapidly, compared to a scenario that only emphasises investment in technology, but not its distribution. Here we assumed that *6_distribution* is negatively correlated to the percentage of rural population (derived from the urban share projections from 2010-2100 NCAR, v9_130115; SSP-Database, 2015) on the basis that a larger share of rural population implies more small scale farming with simple technologies and lower yields. Additionally, the variables in Table A2 were added during the implementation of LPJ-GUESS driven yield development in PLUM.

**Table A2: Yield related variables in PLUM.**

| Variable type | Variable name | Source |
|---|---|---|
| Initial value | *cerealYieldFAO_0* | FAOSTAT, 2015 |
| Initial value | *yieldA_0* | First year of yieldA_t (currently year 2000) |
| Initial value | *q_0* | q_t at time 0, i.e., year 2000 in this version |
| Time series | *yieldP_t* | LPJ-GUESS, potential yield |
| Time series | *yieldA_t* | LPJ-GUESS, actual yield |
| Time series | *gdpPc_t* | Income growth, GDP per capita, SSP data |
| Time series | *urbanShare_t* | Share of population living in urban areas, SSP data |
| Model parameter | *boundaryShare* | Share of cerealYieldFAO_0 that yield can decrease to as a minimum |

Cereal yield *cerealYieldC* is calculated in the following way:

*cerealYieldC* = if *cerealYield\*shareYield* > *cerealYieldFAO_i\*boundaryShare* then *cerealYield\*shareYield* else *cerealYieldFAO_i\*boundaryShare*

To avoid that yields are decreasing to 50% (corresponding to default *boundaryShare* of 0.5) or less than the initial FAO value (*cerealYieldFAO_i*) *cerealYieldC* is kept constant in this case. The LPJ-GUESS based calculated yield (*cerealYield*) is normalised with FAOSTAT cereal yield by multiplying with *shareYield*.

*shareYield = cerealYieldFAO_i/yieldA_i*

*cerealYield* is calculated:

*cerealYield = yieldP_t \*(1 - yieldGap)*

*yieldGap* is calculated:

$yieldGap = 1 - (yieldA\_t * k\_t / (yieldP\_t))$

The function $k\_t$ determines how much of actual yield vs. potential yield is produced during time:

If q < 0.98 then

$k\_t = kmax\_t * (1 - (h\_t * q\_t))$

else $k\_t = kmax\_t(1 - (h\_t * 0.98)$

The maxium value of $kmax\_t$:

$kmax\_t = yieldP\_t / yieldA\_FAO$

The function $h\_t$:

$h\_t = (1 - (1 / kmax\_t)) / q\_i$

The function $q\_t$ describes the impact of investments in technology and distribution of technology on yields. Investments in technology are here assumed to be dependent on income growth (GDP per capita) and distribution of technology is assumed to be related to the share of urban population on total population.

$q\_t = exp(-technol\_t - investment\_t * gdpPc\_t + distribution\_t * (100 - urbanShare\_t))$

The functions $technol\_t$, $distribution\_t$ and $investment\_t$ allow the change of the initial factors which were found by

regression analysis based on statistical analysis of data for the year 1995-2005:

$technol\_t = 0.77 + 6\_technology / 100 * time()$

$investment\_t = (1.80 + 6\_investment / 100 * time()) * 10^{-5}$

$distribution\_t = (2.55 - 6\_distribution / 100 * time()) * 10^{-3}$

The scenario dependent input parameters $6\_technology$, $6\_investment$ and $6\_distribution$ were parameterised guided by their

standard deviations of 0.125, $0.650 * 10^{-5}$ and $1.000 * 10^{-3}$ respectively (see Table B1).

**Appendix B: Input to the conditional probabilistic approach**

**B1 Extended SSP narratives**

**SSP1: Sustainability – Taking the green road**

**Keywords:** Cooperative countries, environmentally friendly, functioning markets

Facilitating governance and institutional structures, cooperative countries, environmentally conscious societies and decreased inequalities contribute in this SSP to the progress towards a sustainable world, including lower population growth. Due to effective international institutions and good information flow between markets, governments and farmers and functioning global markets, agricultural areas are decreased rapidly in case of food overproduction. Countries rely on regional trade and food stocks are tried to be kept low in order to be resource efficient. The conversion of natural land to new

cropland is well regulated in most countries to avoid substantial deforestation and biodiversity loss. Investments in agriculture and agricultural research are staying high in high income countries, and local, context dependent agricultural best management practices (including non-conventional practices as e.g. no-tillage) are implemented in most countries. The

investment in technology continues to result in more efficient animal protein production. An additional important factor for globally increasing yields is the technology transfer between countries and income levels. Equity and education are important in this scenario and contribute to yield improvements as well. The awareness for resource efficiency also decreases food waste and the consumption of refined products, which leads to a decreasing cereal consumption. The environmental

awareness of consumers leads to a slowing down and an eventual decrease in the consumption of dairy and meat products. However, low income countries moderately increase their animal product consumption until they reach consumption levels that are common among Western countries. Environmental degradation is slowing down and the status of land improved, thanks to increasing practice of holistic and sustainable management and afforestation programs.

**SSP2: Middle of the road**

**Keywords:** Business as usual,

In SSP2 trends observed during recent decades continue, including some reductions in resource intensity, but mostly remaining large inequalities between countries and economies. Technological development is moderate and preliminary concentrated to high income countries. Due to limited technology transfer, low income countries do not benefit from advances in agricultural management and yields remain rather low. Agricultural markets are partially functioning and

globally connected, but trade barriers are only reduced slowly. Some countries with limited access to global markets focus more strongly on increased domestic production and self-sufficiency. In general food stocks are hold at moderate levels and the abandonment of cereal land remains unregulated. For new cropland generation, high income countries follow existing regulations, while in some low income countries with rich natural resources unregulated deforestation for cropland generation continuous to be a problem. Environmental degradation continues at historical rates, as no serious efforts are

made to achieve large scale sustainable land management. Additionally the continuing increasing demand for animal products contributes to expansion and intensifying agriculture with some negative environmental impacts.

**SSP3: Regional Rivalry – A rocky road**

**Keywords:** World regions, security of regions, no progress in technologies

In the fragmented world regional blocks are forming, with little international cooperation and protectionist policies of

regions as a result. This leads to little reduction of land intensity, low technological development and generally slow economic growth, but high population growth. However, in some areas wealth is moderately increasing, and so is technological development. The increasing efforts of regions to be more food self-sufficient reduce agricultural trade and increase the food overproduction within regions to ensure sufficient food supply in case of regional harvest shortcomings. Consequently, agricultural area is only abandoned in very slow pace, even if a region is food sufficient. At the same time

weak governance and institutional structures do not provide any strong regulations reducing the conversion of natural land to cropland. Forests and natural grasslands are converted into cropland a larger rates to ensure regional food security. Food consumption, and in particular the consumption of animal products, is continuing to increase in most regions, but at a slower pace for low income countries. The increased demand for food and the non-regulated land use change result in serious environmental degradation.

**SSP4: Inequality – A road divided**

**Keywords:** The few wealthy control, the rest struggles

This world is characterised by high inequality, within and across countries, as well as between economies. In all countries, including low income countries, few very privileged people steer all political, economic and industrial activities. This includes agriculture, which is strongly divided into highly industrialised large scale monoculture agriculture steered by the privileged and small scale farming performed by a large group of poor people. Investments in agricultural development of the industrial agriculture are large, but no technology transfer occurs to the small scale farming and here yields remain low. If the industrialised agricultural production is not profitable, cropland is abandoned at large rates, while at the same time natural land is converted in large rates to new cropland without considering environmental and social effects. The absence from sustainability regulations leads to serious environmental degradation, affecting the poor and making them even more vulnerable. The global food trade is dominated by the industrial agricultural businesses with very limited access for small scale farmers. Small scale farmers therefore rely more on self-sufficient agricultural systems. While the privileged society increases it's consumption of animal products, the large group of poor people cannot afford large increases in meat and dairy consumption. The overall demand for food production is therefore not proportionally increasing with the high population growth, as most of the world's people cannot afford an expensive diet in times of economic uncertainty.

**SSP5: Fossil-fuelled development – Taking the highway**

**Keywords**: Resource intensive, no compromises to gain material wealth

In this world economic, resource intensive development is prioritised and while this leads to eradication of extreme poverty, is comes at environmental costs. Developing countries are pushed in their development and soon all countries share a resource intensive lifestyle, including high levels of animal products consumption. The high demand for these and other agricultural product is fulfilled by highly engineered agricultural systems. Investments into agricultural technology are very high. Increasing agricultural specialisation of countries is common too, however, often connected to very resource intensive production, both, in terms of water and fertilizers. Agricultural area is also expanding into natural areas at larger rates if necessary. Solutions to environmental problems are not tackling the problem's roots, but only its symptoms. However, the global food market is well functioning and keeps the total food stocks decrease slowly.

**B2 Examples of rationality for changes of trend and uncertainty levels for PLUM input parameters conditional on SSPs**

An example of the importance of being explicit about baseline trends is the following: in SSP5 the very strong trend of technology improving agricultural management (Table 1, SSP5, t*echnology*: +++) is, when compared to the generally strong baseline trend of *technology* : ++, not considered extreme. To illustrate this approach further, consider for instance the scenario element "globalization" from Table 2, which we assumed will influence, jointly with scenario elements "international trade" and "agriculture", the input parameter that guides the level of food production in the model. If we take SSP3, a degree of de-globalisation and enhanced regional security is assumed to be taking place in future (O'Neill et al., in

press). Consistent with a world where the trend of globalization is reversed and regional security is important, we assumed for SSP3 that production levels would be higher (+++, Table 1) compared to the current trend in order to ensure supply of demand internally. Vice versa for SSP1, where "globalization" leads to "connected markets, regional production" (O'Neill et al., in press) we assumed that production levels would be lower (--, Table 1) than present-day levels since food would be
distributed more efficiently around the globe. The reduction of production levels would also decrease food waste, which is consistent with SSP1's "policy orientation" towards sustainable development.

**B3 Quantitative values for input parameters**

For each scenario and each input parameter quantitative values were derived by sampling from Table B1 based on the scenario and input parameter's qualitative notions in Table 1.
This matrix (Table B1) was populated by first placing the (baseline) mean value (Engström et al., 2016) based on the quantitatively estimated baseline trend within the matrix. Secondly, we identified minimum and maximum values for each input parameter, based on statistical analysis of historic data or the authors' judgement as described in Engström et al (2016). Thirdly, these values informed the extremes (--- and +++) and the entries between the extremes and the baseline mean values were filled with evenly interpolated values. The quantitative values for the qualitative uncertainty levels low, medium and
high were informed by variability in historic data. We assumed the historic standard deviations to be generally high because data were analysed over a time period (1961-1990) where substantial changes in consumption and production patterns led to high heterogeneity in the data (Alexander et al., 2015). We used the historic standard deviation for the high uncertainty value. The medium and the low uncertainty values are two thirds, and one thirds of the high uncertainty level respectively (Table B1).
The values for change in trends (mean) and uncertainty value (standard deviation) were used to create the probability distribution function for each input parameter (PDFs). We assumed normal distribution for all input parameters, except parameters 7-9. These parameters are maximum values and it seemed more plausible that they are equally likely (uniform distribution). For population we truncated the distribution, because the very low population projections of SSP1 (peak of global population size in 2050-2055 at 8.5 billion, a decline in total population size to 6.9 billion in 2100) requires very
stringent decline in fertility rates, and even for SSP1 with high focus on education (the most important driver for changes fertility rates) it seemed very unlikely to us that the projections will lower minus one standard deviation. With the reversed argumentation we truncated upper bound for the PDF for population for SSP3, as SSP3 population projections derive at a very high total population size in 2100 of 12.6 billion people.

**Table B1: Matrix with quantitative values for changes in trend from --- to +++ (mean values), and the uncertainty levels low, medium and high (1 Standard Deviation, SD). Values are based on analysis of historical data, except values with * were estimated by the author (see Engström et al., 2016). # indicates that values are maximum values rather than standard deviations. Grey cells are baseline values, corresponding to baseline trend in Table1.**

| Nr | Input parameter (unit) | Change in trend, mean values | | | | | | | Uncertainty, 1 SD | | |
|---|---|---|---|---|---|---|---|---|---|---|---|
| | | --- | -- | - | 0 | + | ++ | +++ | low | medium | high |
| 0 | gdpVar (%) | country level time series for each SSP | | | | | | | calculated based on other projections, see Table 1 | | |
| 0 | popVar (%) | | | | | | | | | | |
| 1 | overProdRate (1/time) | -0.004 | -0.003 | -0.002 | -0.001 | 0 | 0.001 | 0.002 | 0.0002 | 0.0004 | 0.0006 |
| 2 | cerealVar (1/time) | -0.003 | -0.002 | -0.001 | 0 | 0.001 | 0.002 | 0.003 | 0.0002 | 0.0004 | 0.0006 |
| 3 | meat 1 (kg meat per capita/log(GDP per capita)) | -10 | -5 | 0 | 5 | 10 | 15 | 20 | 2 | 4 | 6 |
| 3 | meat 2 (kg meat per capita/log(GDP per capita)) | -6 | -3 | 1 | 3 | 5 | 10 | 15 | 1 | 2 | 3 |
| 3 | meat 3 (kg meat per capita/log(GDP per capita)) | -5 | 0 | 5 | 10 | 15 | 20 | 25 | 1 | 2 | 3 |
| 3 | meat 4 (kg meat per capita/log(GDP per capita)) | -2.5 | 0 | 2.5 | 5 | 10 | 15 | 20 | 1 | 2 | 3 |
| 4 | milk 1 (kg milk per capita/log(GDP per capita)) | -10 | -5 | 0 | 5 | 10 | 15 | 20 | 2 | 4 | 6 |
| 4 | milk 2 (kg milk per capita/log(GDP per capita)) | -4 | -2 | 0 | 2 | 6 | 10 | 14 | 1 | 2 | 3 |
| 4 | milk 3 (kg milk per capita/log(GDP per capita)) | -5 | 0 | 5 | 10 | 15 | 20 | 25 | 2 | 4 | 6 |
| 4 | milk 4 (kg milk per capita/log(GDP per capita)) | -2.5 | 0 | 2.5 | 5 | 10 | 15 | 20 | 2 | 4 | 6 |
| 5 | fcrImp* (1/time) | -0.002 | -0.001 | 0 | 0.001 | 0.002 | 0.003 | 0.004 | 0.0003 | 0.0006 | 0.0009 |
| 6 | distribution (1/time) | -1 | -0.66 | -0.33 | 0 | 0.33 | 0.66 | 1 | 0.15 | 0.3 | 0.45 |
| 6 | technology (1/time) | -0.125 | -0.08 | -0.04 | 0 | 0.04 | 0.08 | 0.125 | 0.03 | 0.06 | 0.09 |
| 6 | investments (1/GDP per capita *time) | -0.65 | -0.43 | -0.21 | 0 | 0.21 | 0.43 | 0.65 | 0.1 | 0.2 | 0.3 |
| 7 | abandonCL# (unitless) | 0.01 | 0.015 | 0.02 | 0.023 | 0.04 | 0.055 | 0.07 | 0.003 | 0.006 | 0.009 |
| 7 | abandonCL_D# (unitless) | 0.01 | 0.011 | 0.013 | 0.015 | 0.025 | 0.035 | 0.05 | 0.003 | 0.006 | 0.009 |
| 8 | newCL# (unitless) | 0.01 | 0.011 | 0.013 | 0.015 | 0.25 | 0.035 | 0.048 | 0.003 | 0.006 | 0.009 |
| 8 | newCL_D# (unitless) | 0.01 | 0.018 | 0.025 | 0.029 | 0.035 | 0.04 | 0.045 | 0.003 | 0.006 | 0.009 |
| 9 | newCLs# (unitless) | 0 | 0 | 0 | 0 | 0.015 | 0.03 | 0.048 | 0.003 | 0.006 | 0.009 |
| 9 | newCLs_D# (unitless) | 0.01 | 0.018 | 0.025 | 0.029 | 0.035 | 0.04 | 0.045 | 0.003 | 0.006 | 0.009 |
| 10 | grassForest* (unitless) | 0.5 | 0.5 | 0.5 | 0.5 | 0.5 | 0.5 | 0.5 | 0.03 | 0.06 | 0.09 |
| 11 | residualNV* (%) | 4.0 | 6.0 | 8.0 | 10.0 | 12.0 | 14.0 | 16.0 | 1.0 | 2.0 | 3.0 |
| 12 | croplandDeg* (1/time) | 0.04 | 0.06 | 0.08 | 0.10 | 0.12 | 0.14 | 0.16 | 0.03 | 0.06 | 0.09 |
| 12 | forestDeg* (1/time) | 0.004 | 0.006 | 0.008 | 0.010 | 0.012 | 0.014 | 0.016 | 0.003 | 0.006 | 0.009 |

**Appendix C: Additional output**

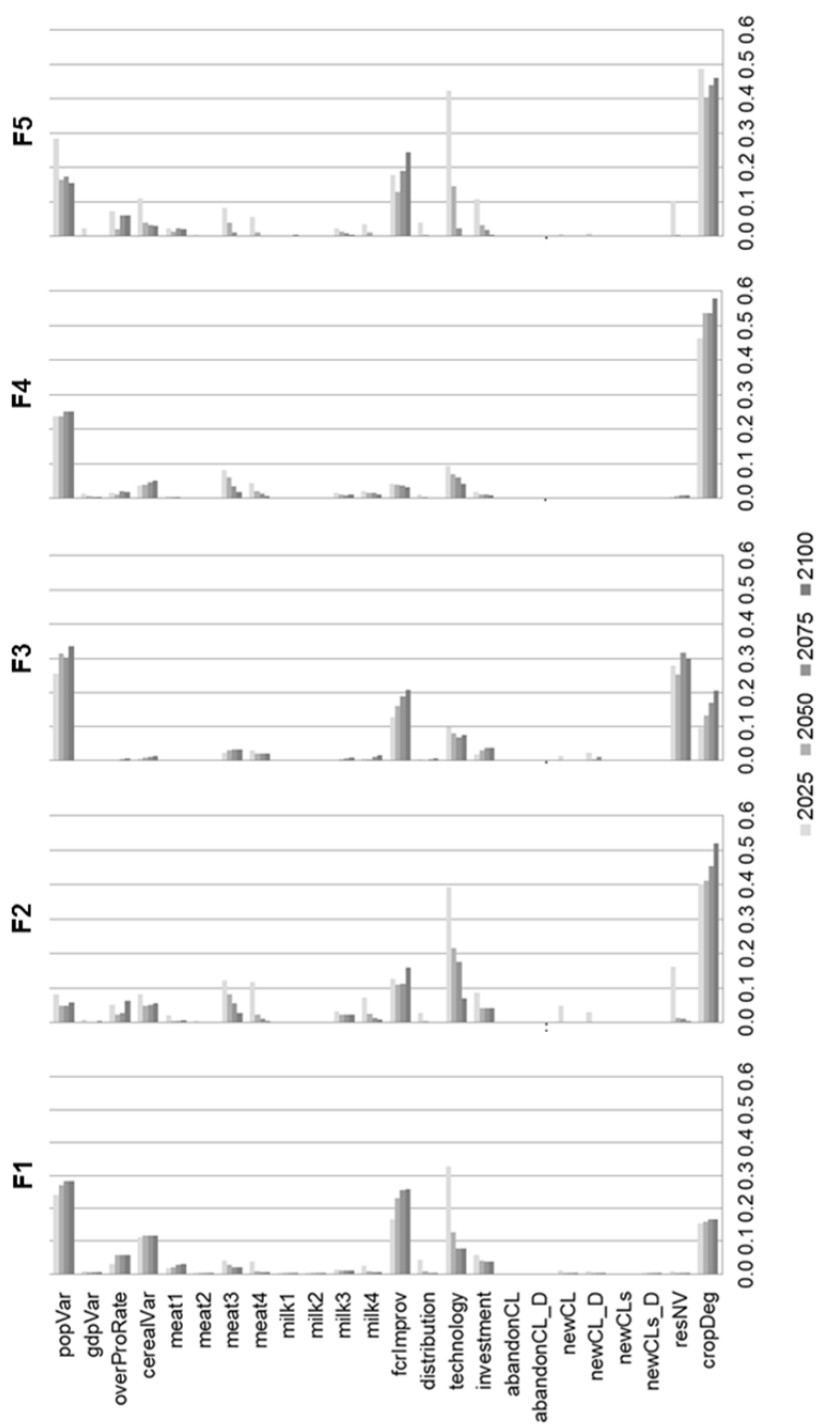

**Figure C1: Total importance of cropland to uncertainty of input parameters conditional to SSPs.**

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

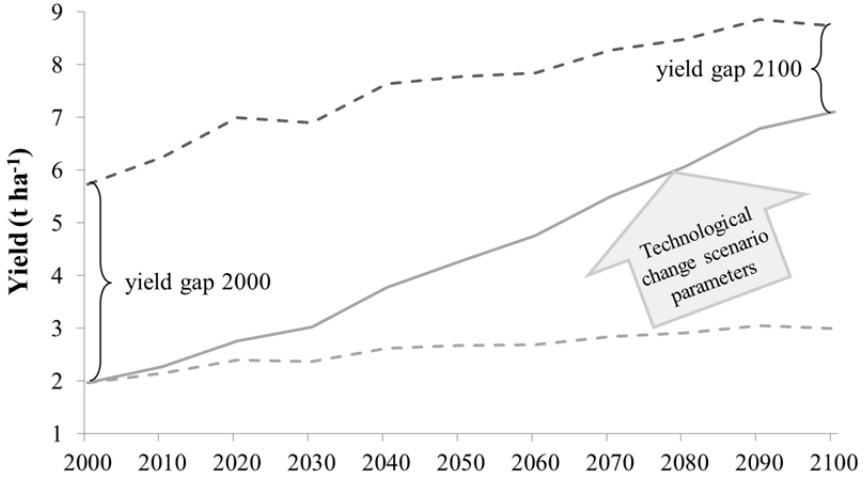

**Figure 1: Decreasing yield gap for Ukraine. The scenario parameters related to technological change determine how rapidly the**
15  **yield gap decreases over time (the arrow only being symbolic, indicating the drivers of changing yield gap).**

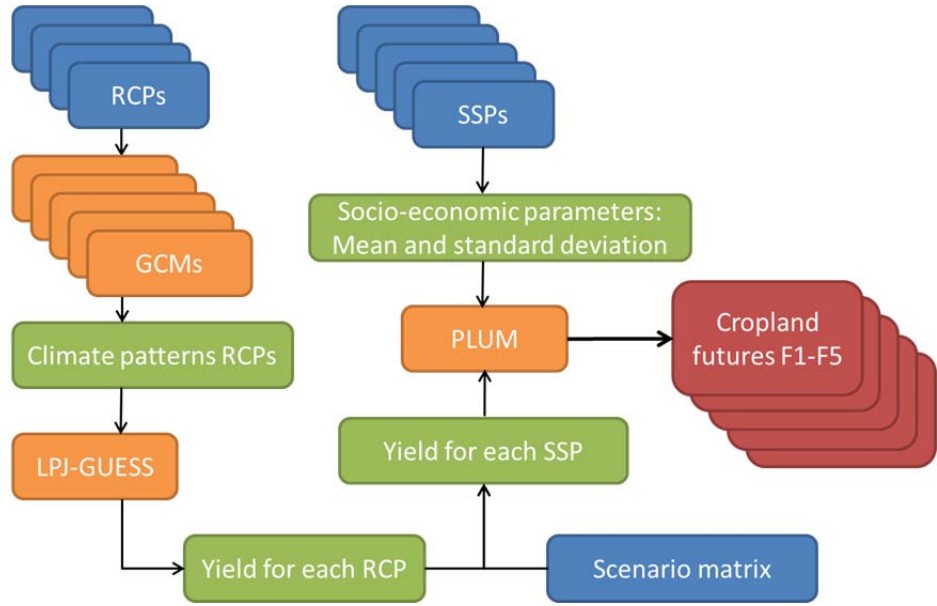

**Figure 2: The SSP-RCP modelling framework. The RCPs and SSPs, as well as the scenario matrix (author judgement about the distribution of RCPs conditional on SSPs) are input to the model (indicated in blue). Models (indicated in orange, GCMs: General Circulation Models) use input or results of other models (intermediate results, indicated in green). The final outputs of the modelling framework are the cropland futures F1-F5 (indicated in red).**

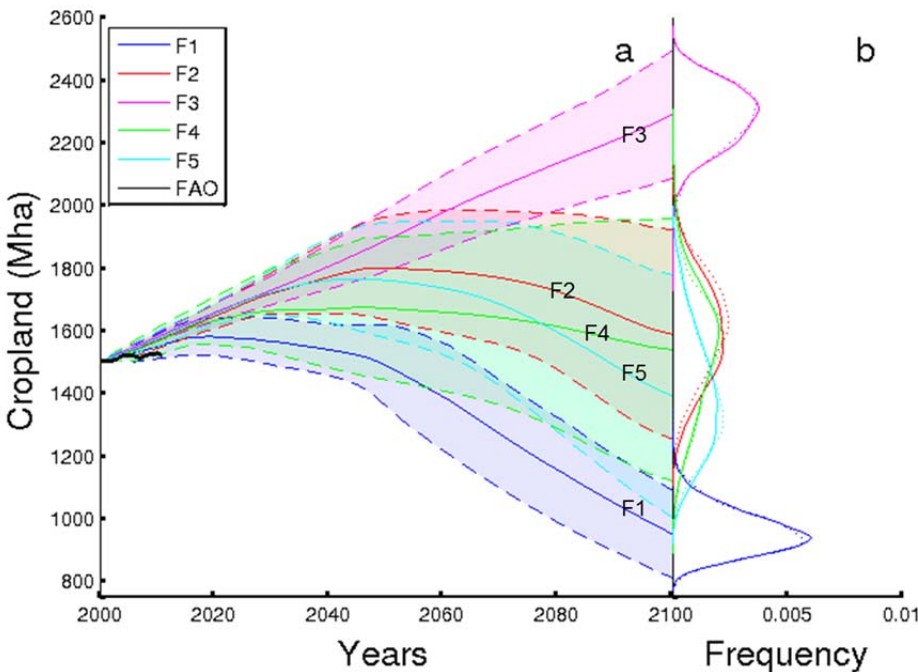

**Figure 3: Panel a: cropland development from 2000 to 2100 for the five cropland futures (solid line: mean, range with dashed lines: +/- 2 standard deviations). Panel b: PDFs fitted to all runs for each scenario in 2100, solid lines are runs with sampling yield variations due to GCM patterns and the scenario matrix and dashed lines are runs where the mean yield was used.**

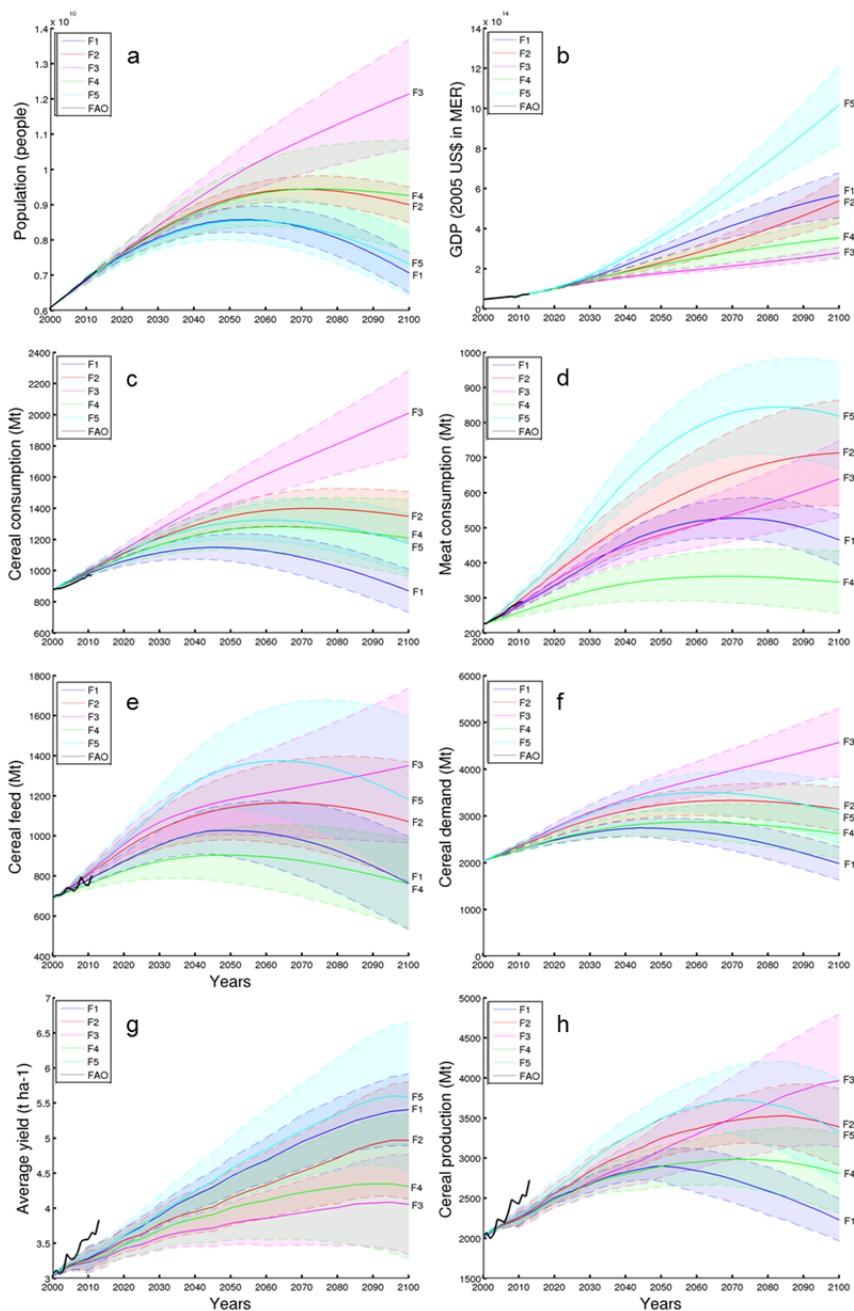

**Figure 4: Global population, GDP, cereal consumption, meat consumption, cereal feed, cereal demand, mean cereal yield and cereal production for the five scenarios F1-F5. Solid lines indicate the scenario mean, dashed lines indicate the range based on the mean ± two standard deviations.**

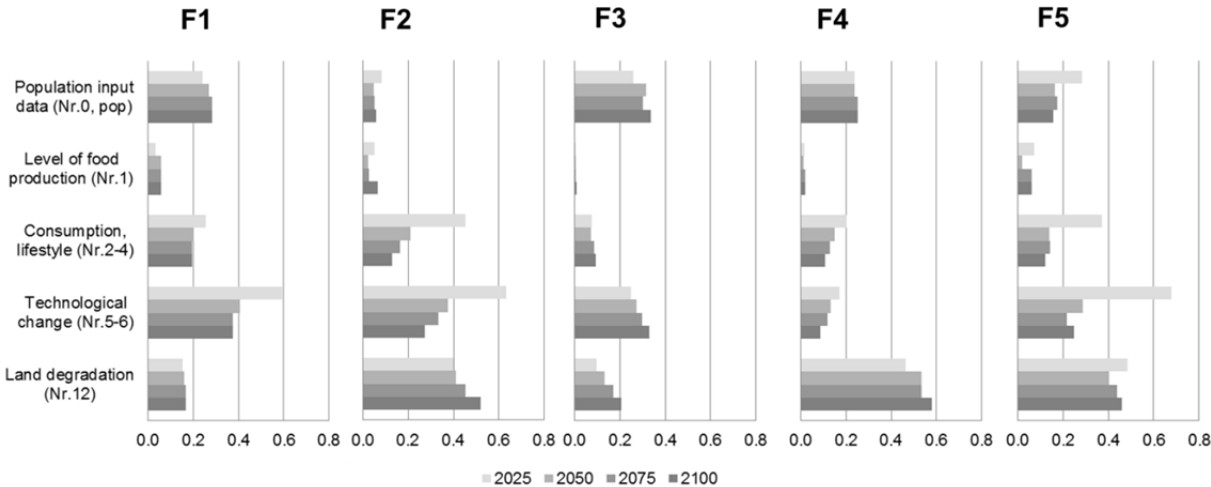

**Figure 5: Total importance of global cropland for the futures F1-F5 to uncertainty of input parameters, aggregated to input parameter groups as in Table 2 (for non-aggregated results see Fig. C1 in Appendix C).**

**Table 1: Quantitative uncertainty levels (uncert.) for *gpd* and *pop* and qualitative estimates for the PLUM input parameters (Nr. 1-12) for changes in trend until 2100 relative to estimated current trends and uncertainty levels. Nr. indicates the parameter's membership to the parameter-group (see Table 2). Quantitative values for parameters 1-12 were sampled from Table B.1 in Appendix B.**

| Nr. | Input parameter explanation | Input parameter | baseline trend | SSP1 trend | SSP1 uncert. | SSP2 trend | SSP2 uncert. | SSP3 trend | SSP3 uncert. | SSP4 trend | SSP4 uncert. | SSP5 trend | SSP5 uncert. |
|---|---|---|---|---|---|---|---|---|---|---|---|---|---|
| 0 | economic income | gdp | 0 | ++ | 7.0% | 0 | 7.5% | -- | 3.5% | 0 | 12.5% | ++ | 7.0% |
| 0 | population | pop | 0 | -- | 4.0% | 0 | 2.0% | ++ | 6.0% | 0 | 6.0% | -- | 4.0% |
| 1 | overproduction rate | overProdRate | + | -- | medium | + | high | +++ | medium | ++ | high | 0 | high |
| 2 | cereal variability | cerealVar | 0 | - | medium | + | medium | ++ | low | 0 | medium | ++ | medium |
| 3 | slope of meat consumption function for country groups, 1: traditionally high meat consuming countries, 2: traditionally low meat consuming countries, 3: transitioning countries, 4: developing countries | meat 1 | + | -- | medium | + | medium | ++ | low | ++ | medium | +++ | high |
| 3 | | meat 2 | + | - | medium | + | medium | ++ | low | + | medium | +++ | high |
| 3 | | meat 3 | + | + | low | ++ | high | ++ | medium | - | high | +++ | medium |
| 3 | | meat 4 | 0 | + | low | ++ | high | + | medium | -- | high | +++ | medium |
| 4 | slope of milk consumption function for country groups, 1: traditionally high meat consuming countries, 2: traditionally low meat consuming countries, 3: transitioning countries, 4: developing countries | milk 1 | 0 | -- | medium | + | medium | ++ | low | ++ | medium | +++ | high |
| 4 | | milk 2 | 0 | - | medium | + | medium | ++ | low | + | medium | +++ | high |
| 4 | | milk 3 | ++ | + | low | ++ | high | ++ | medium | - | high | +++ | medium |
| 4 | | milk 4 | + | + | low | ++ | high | + | medium | -- | high | +++ | medium |
| 5 | feed conversion rate improvement | fcrImp | ++ | ++ | medium | ++ | medium | - | high | + | medium | +++ | medium |
| 6 | distribution of technology | distribution | + | +++ | high | + | medium | -- | medium | --- | high | ++ | medium |
| 6 | technological change | technology | ++ | ++ | medium | + | high | - | medium | + | medium | +++ | high |
| 6 | investments in technology | investments | ++ | ++ | medium | ++ | high | - | medium | + | medium | +++ | high |
| 7 | abandonment rate of cropland, D = developing countries | abandonCL | 0 | +++ | medium | 0 | low | - | medium | + | high | ++ | low |
| 7 | | abandonCL_D | 0 | +++ | medium | 0 | low | - | medium | + | high | ++ | medium |
| 8 | conversion rate of new cropland, D = developing countries | newCL | 0 | - | medium | 0 | medium | + | medium | ++ | high | ++ | low |
| 8 | | newCL_D | 0 | - | medium | 0 | medium | + | medium | +++ | high | ++ | medium |
| 9 | conversion rate of new cropland for self-sufficiency, D = developing countries | newCLs | 0 | 0 | na | 0 | na | ++ | medium | + | high | -- | low |
| 9 | | newCLs_D | 0 | - | low | + | medium | +++ | high | ++ | high | -- | low |
| 10 | ratio of cropland converted to/from grasslands and forests | grassForest | 0 | 0 | low | 0 | medium | 0 | high | 0 | high | 0 | medium |
| 11 | residual natural vegetation | residualNV | 0 | ++ | low | 0 | medium | --- | medium | -- | medium | - | medium |
| 12 | cropland degradation | croplandDeg | 0 | -- | medium | 0 | medium | ++ | low | + | high | ++ | medium |
| 12 | forest degradation | forestDeg | 0 | -- | medium | 0 | medium | ++ | low | + | high | ++ | medium |

**Table 2: PLUM input parameter groups and the influence of *SSP scenario elements* (from O'Neill et al., in press). In PLUM several of the input parameters were grouped together conceptually, as indicated by the parameter group number (Nr). For example, there are four input parameters that describe meat-consumption trajectories of different income and cultural groups (meat1-meat4, see Table 1), which all belong to meat consumption, parameter group Nr. 3.**

| Nr | PLUM input parameter groups | Influencing *scenario elements* (O'Neill et al., in press) |
|----|------------------------------|------------------------------------------------------------|
| 1 | Level of food production | International trade, globalization, international cooperation, environmental policy, policy orientation, institutions, agriculture |
| 2 | Cereal consumption | Consumption & diet |
| 3 | Meat consumption | Inequality, consumption & diet |
| 4 | Milk consumption | Inequality, consumption & diet |
| 5 | Efficiency of animal production | Technology development, agriculture |
| 6 | Technological change for yield | Technology development and transfer, agriculture |
| 7 | Cropland reduction | Land use |
| 8 | Cropland expansion | Institutions, land use |
| 9 | Cropland for self-sufficiency | International trade, globalization, land use |
| 10 | Grassland vs. Forest | Environmental policy |
| 11 | Remaining natural vegetation | Environmental policy, policy orientation |
| 12 | Land degradation | Environmental policy |

**Table 3: Conditional probabilities (ranging from very low to very high) of SSPs resulting in RCPs based on authors' judgement.**

|       | RCP 2.6 | RCP 4.5 | RCP 6 | RCP 8.5 |
|-------|---------|---------|-------|---------|
| **SSP1** | very low | medium | medium | 0 |
| **SSP2** | 0 | very low | high | low |
| **SSP3** | 0 | low | high | medium |
| **SSP4** | 0 | medium | high | very low |
| **SSP5** | 0 | very low | medium | high |

**Table 4: Scenario-matrix translated to quantitative probabilities.**

|          | RCP 2.6 | RCP 4.5 | RCP 6  | RCP 8.5 | Sum |
|----------|---------|---------|--------|---------|-----|
| **SSP1** | 0.0909  | 0.4545  | 0.4545 | 0.0000  | 1   |
| **SSP2** | 0.0000  | 0.0909  | 0.6818 | 0.2273  | 1   |
| **SSP3** | 0.0000  | 0.1667  | 0.5000 | 0.3333  | 1   |
| **SSP4** | 0.0000  | 0.3704  | 0.5556 | 0.0741  | 1   |
| **SSP5** | 0.0000  | 0.0741  | 0.3704 | 0.5556  | 1   |