# Peer review of "Assessing uncertainties in global cropland futures using a conditional probabilistic modelling framework"

_Earth System Dynamics, 2016_

## Referee Comment (RC1) · Anonymous Referee #1 · 9 May 2016

Dear authors,

you present an interesting manuscript deriving probability distributions for global cropland areas from the new narratives describing the IPCC SSP scenarios. I think this is a valuable contribution as it tries to associate numbers with the narratives, helps imagining a bit more vividly how the world could look in these scenarios and what implications this could have for land use, and also serve as an example for other researchers that work with these scenarios. At the same time it exposes the uncertainties left open by the scenarios and the difficulties in working with them.

In general, I find the methodology you used and the results that you produced defensible, although I miss a more explicit discussion of some of the limitations of the analysis

and its relationship to more structured land use models. Some thoughts on this:

a) You write that PLUM is consistent with other, more complex models of global land use change. (Quotation still under review) Besides the fact that this may just mean that it inherited their problems, what is more important here is the question in how far the results you obtain in this article are representative of the results you'd obtain with these models. Does the fact that PLUM has a simpler structure mean that the outcome distributions are wider than with other models, because there is less structure constraining them? Or are there locations in the outcome distribution that may not be produced by PLUM (but by other models), because it would require interaction effects between input variables/processes that are not incoporated in PLUM? In the end, are other models likely to produce the same outcome distributions (or wider or narrower ones) if they could be used in the same experiment?

b) That PLUM reproduces land use change between 1990-2010 counts in its favor, but cannot really be judged without the specific assumptions made in the validation runs, and does not guarantee much for land use change on longer time scales in a potentially very different environment in which fundamental parameters may change. What needs to be considered here is whether the structure assumed within PLUM is likely to remain unchanged and potential structural changes can explicitly captured within the input parameters. If that is the case, the comparatively reduced structure of PLUM may be an asset that may be exposed more clearly.

My other concern is mainly with the exposition of your research in the manuscript. It lacks a bit of detail on some important assumptions and is unclear on others. My specific remarks in this respect:

1. section 1,2: I think the understandability and readability of your article would benefit from restructuring. You should mention earlier and consistently from the beginning that you are going to compare five future scenarios which correspond to the five SSPs. The probability distributions/ranges assumed for uncertain parameters - including the RCPs - are depending on the scenario and differ between scenarios. You might also want to consider reordering subsections in the Methodology section (2), starting with a discussion of the scenario/uncertainty approach (currently 2.2) and only then give details on the PLUM model (currently 2.1). This would proceed from the more general to the more specific and follow the structuring of the introduction. In section 2.2 (original numbering), you should then start with something like. "We construct five conditional probabilistic futures (F1-F5), each constructed based on the qualitative and quantitative information in the corresponding SSP (SSP1-5). ... The extent of climatic change within each future was determined by assigning a SSP-specific probability ot each RCP ... " or something similar. I think this would help readers grasp what you are doing at the first reading and right from the start and not having to re-read a few times until that connection becomes clear. There is a clear hierarchy between scenario and other input (and RCPs are just one of them) in your scenario construction and your description should reflect that.

2. p.2, l.26: I think this sentence is misleading, because basically you use the five SSP scenarios to construct futures. The RCPs are modeled as dependent on SSP, in the same way as many other uncertain input factors. I think it would be helpful for the understanding of the article to make that clear right from the start and introduce the F1-F5 already here.

3. p.3, l.9: A constant cropland-cereal land ratio seems a plausible assumption for 1990-2010, but is there any indication that this will hold for longer time-scales and everywhere around the world?

4. p.3, l. 22: It is not clear to me, how actual and potential yields were calculated: You write that the potential yield is the yearly maximum in each grid cell. Does that mean the maximum yield simulated in any year between 2000 and 2100? Or, for each year, the maximum that was obtained for any cell in the country?

(Be clear what is per cell and what is per country here and elsewhere.) Likewise, what is the difference to the actual yield? Is the actual yield also simulated or is it observed in the MIRCA dataset directly? The paragraph is not very clear.

5. p.3, l.24 and l. 30: You seem to do a bias correction here, even twice: Once in LPJ-GUESS and once in PLUM, making simulated yields match observed yields. I miss an explanation on why this is necessary and justified, as well as a discussion on which implications this has for the conclusions that can be drawn from your model. (Is it still applicable for extrapolation into the future, i.e. out-of-sample?) (see Ehret et al. 2012, Hydrology and Earth System Sciences, albeit for a different context)

6. p.5, l.31 : Strictly speaking, you are referring to coefficients of variation rather than standard deviations as you are using percentages, aren't you? Or do I misunderstand the paragraph.

7. p.7 l.3; p.7 l.10; p.5, l.17: Although based on established secondary data, there still seems to be quite an "ad hoc" component in the specification of probability distributions for parameters: Is there a justification for assuming normal distributions for all parameters? Is there a justification for assuming independence between input factors. One might argue that uniform or heavy-tailed distributions might better reflect the uncertainty about these distribution themselves and one might also imagine that some parameters could be covariates. E.g. which population projections are assumed by the studies you consulted for projections of global GDP development? Are certain population projections correlated with lower or higher GDP development? Independent sampling from normal distributions for many parameters means that cases where several parameters are at their extremes are unlikely to occur.

8. p.7 l.15 : Did you consider to use an alternative sampling schemes (importance sampling, Sobol' sequences etc.) to increase sampling efficiency?

9. p.8. l.21ff.: As the description of yield simulations in section 2.1 left me in some doubts, I cannot really follow at this point either. The paragraph here is not clear about your intentions nor the details of implementation in using SVD for analysis and resampling. What "pattern resulting from the GCM-RCP yield projections" is to be preserved? Is it the differences between countries? The difference between GCMs? The differences between RCPs? All in all, you do not explain yourself well here...

10. p.8. l.24f.: Isn't it SINGULAR value decomposition rather than single value decomposition? Equation 1 is pretty cryptic. Can you use SVD in a formula and add to it? Isn't the outcome of SVD three matrices? Which one is used in this equation here? (Is there any reference for this use of SVD or did you come up with this approach yourselves?)

11. p.8. l. 32 (Eq.2): A bit more space between the comma and the second epsilon would greatly increase readability.

12. p.9. l.14: There is a bit of a confusion about references here. What is the Sobol-jansen method? Is it an R implementation of the method described in Lilburne and Tarantola 2009 or Saltelli 2010 or Engström 2016? Or is it a specific method of GSA? Please clarify and elaborate a bit more.

13. sections 3.2, 4 and 5: I found your use of convergent and divergent quite confusing. You use it to refer to the fact that three scenarios (F2,F4,F5) have strongly overlapping domains of the outcome distribution, which you describe as convergent, while F1 and F5 are clearly distinct from the each other and from the other three, which you describe as divergent. But, on the other hand, F1 and F3 show rather peaked distributions, which one might also associate with convergence (uncertainty within the scenario converges around a common mean) and which would be closer (although also not matching) the common use of convergence in statistics, i.e. the convergence to a value as repetitions increase. (F2-F4 show

rather wide distributions in contrast that would then rather be divergent. ) Especially, when you speak of a convergent (resp. divergent) scenario, at least I rather have the association of convergence (divergence) WITHIN the scenario and not between several scenarios.

At least to me, this caused some confusion, especially since you state "the confidence in the model outcomes for F1 and F3 are the highest, DESPITE the fact that these two scenarios show divergent global cropland development." If you meant convergence/divergence in the statistical sense, this contrast between confidence and divergence would warrant the despite. However, divergence in your use, i.e. the observation that certain scenario assumptions lead to other outcomes than the (slight) majority of other scenario assumptions is not necessarily at odds with higher confidence.

14. I would avoid the use of convergence/divergence for the pattern you describe. E.g. you could maybe use "strong similarity" instead of "strong convergence" in p.10 l.14, and "represent the extremes of simulated" instead of "show divergent" on p.10,l.10. On p. 11, l.18, you could simply replace "converge" by what you put in parenthesis anyway, and so on.

15. section 3.3: Why didn't you include the RCPs in Fig.5 (you mention in section 3.1 that their influence is minor, but nevertheless this would complete the picture and put a number to your statement.)

16. sections 2,4: While you do an extensive uncertainty analysis on parameters that capture how the future might develop, you seem to use - in several places - model parameters that are rather ad hoc, without deeper justification, e.g. a cereal-to-cropland share of 60

17. section 5 (Conclusions): I miss two important aspects in your conclusions:

- I miss a short summary of the crucial assumptions incorporated in our model that the obtained results are based on. "Considering current knowledge on crop responses incorporated in crop gorwth model X, a fixed cereal to land share, etc .. , we conclude .. " Especially in uncertainty analysis such as yours, where a lot of parameters are varied, it is easy to loose track of what are the crucial pillars that the analysis is built on and that give structure to the analysis. You do not vary everything, but there are certain assumptions and model structure that are fixed and bound the results that you obtain.

- I also miss a reflection of what we have learned and in how far you were successfull in dissecting the large uncertainty range you metion in the first paragraph of the introduction. Certainly, the uncertainty range was not reduced. So, what have we learned? We know something more about which conditions are necessary to produce the extremes of the distribution. Anything else ...? To what extent is this a new outcome and not yet visible from looking at and disentangling the results of the yet unpublished comparison study (Alexander et al., under review ) you cite?

18. p. 13 l.15f: Again the use of divergent and convergent here is confusing and misleading (see above).

19. Anywhere: What about the Shared Policy Assumptions? Are they reflected anywhere?

20. Fig.4: These graphs - as they are - are only readable in color. I could not distinguish different futures on a grayscale print out. If possible, consider placing a scenario label next to each line, then one could also read these graphs in black-and-white.

---

## Referee Comment (RC2) · Anonymous Referee #1 · 13 May 2016

Unfortunately, something was lost from the remark number 16 in reviewer comment 1. The full remark should read:

sections 2,4: While you do an extensive uncertainty analysis on parameters that capture how the future might develop, you seem to use - in several places - model parameters that are rather ad hoc, without deeper justification, e.g. a cereal-to-cropland share of 60%, the assumption that a higher rural share of population means lower production efficiency, etc. These parameters cannot be considered set in stone, especially since you indiscriminately apply them to every country in the world. Why didn't you include a certain variation of these parameters in your uncertainty analysis?

---

## Referee Comment (RC3) · Anonymous Referee #2 · 1 Jun 2016

Dear authors,

The manuscript provides an interesting insight into the source and magnitude of uncertainty surrounding global crop-land areas under different SSP scenarios. You provide a detailed methodology to quantify uncertainty ranges for various socio-economic parameters. However, I feel the complex methodology is not easy to follow from the way the manuscript is currently structured. My additional comments:

1. Does the LPJ-GUESS model used to derive actual and potential yields contain a representation of crop response to heat stress? Under climate scenarios modelled (particularly RCP8.5) temperature changes in several countries will result in certain crops becoming unviable.

2. Do actual yields from LPJ-GUESS consider the influence of pests and diseases, and how the influence of this may change over time under different climate scenarios?

3. Following from comments 1 and 2, is the LPJ-GUESS model overestimating actual crop yields (and perhaps potential)? Given that you state agricultural land use is highly sensitive to uncertainties in crop yield growth rates how does this impact your results?

4. You state scaling factors have been applied to both LPJ-GUESS yields (P3 L24) and yield calculated by PLUM (P3 L30) but it is unlikely that these factors should be constant through time.

5. The probabilistic futures F1-F5 are not clearly linked to specific SSPs. I also think the manuscript would benefit from a description of each SSP (even if only very brief) within the manuscript in addition to those in the appendix.

6. Whilst the study has considered uncertainty of socio-economic variables in great detail, it's unclear how indirect climate uncertainties have been incorporated within the study, for instance, uncertainty within the yield modelling. The study concludes that uncertainties arising from climate variability do not strongly affect the range of global cropland futures but perhaps uncertainty in climate influences are under-represented in the methodology? For the two sets of monte-carlo simulations 3600 runs have been conducted for purely socio-economic parameter investigation (or 720 per SSP) and in conjunction with the 4 RCPs this number increases to 7200 (or 1440 per SSP). So the sampling across an individual SSP-RCP matrix is sparser in comparison.

7. P13 L3: This sentence is confusing. From the previous description of the methodology it seems simulations under RCP8.5 were conducted based on the probability of this scenario within SSPs. Yet this sentence indicates the potential impact of RCP8.5 is under-represented (links to previous comment).

8. The discussion should explore further why F2, 4 and 5 strongly converge, and why F1 and F3 diverge, linking this to SSP storylines.

[Figure]

9. Fig 4: It is difficult to distinguish between scenarios in this image, even in colour.

---

## Author Comment (AC1) · 25 Jun 2016

We would like to thank both reviewers for their suggestions, and will address these in the revised manuscript. We detail the changes made below.

Response to reviewer 1:

Dear authors, you present an interesting manuscript deriving probability distributions for global crop-land areas from the new narratives describing the IPCC SSP scenarios. I think this is a valuable contribution as it tries to associate numbers with the narratives, helps imagining a bit more vividly how the world could look in these scenarios and what implications this could have for land use, and also serve as an example for

other researchers that work with these scenarios. At the same time it exposes the uncertainties left open by the scenarios and the difficulties in working with them.

Reply: Thank you for these supportive comments.

In general, I find the methodology you used and the results that you produced defensible, although I miss a more explicit discussion of some of the limitations of the analysis and its relationship to more structured land use models. Some thoughts on this:

a) You write that PLUM is consistent with other, more complex models of global land use change. (Quotation still under review) Besides the fact that this may just mean that it inherited their problems, what is more important here is the question in how far the results you obtain in this article are representative of the results you'd obtain with these models. Does the fact that PLUM has a simpler structure mean that the outcome distributions are wider than with other models, because there is less structure constraining them? Or are there locations in the outcome distribution that may not be produced by PLUM (but by other models), because it would require interaction effects between input variables/processes that are not incoporated in PLUM? In the end, are other models likely to produce the same outcome distributions (or wider or narrower ones) if they could be used in the same experiment?

Reply: Even if global land-use models attempt to represent the same system, in practise, different model approaches (e.g. economic general equilibrium models vs. physical supply models), assumptions, or sectorial focus, result in diverse model structures. When these different models are fed with harmonized input data, the results differ (Schmitz et al., 2014). In a recent comparison of a wide range of models and scenarios (based on the same data as the quotation still under review; Alexander et al., under review) that supplied a wide range of cropland outcomes, PLUM output was well within the spread seen in other models (Prestele et al., 2016). At the global aggregated scale, we are confident that PLUM results are representative for changes that could occur in the agricultural system as we know and understand these now. Regarding your

questions how the outcome distributions produced with PLUM would compare to the outcome distributions of more complex models: We would argue that both, wider or narrower, outcomes could occur in principle. The relatively simple model structure in PLUM, allows externalising all global parameters (but not e.g. country-specific shares of cereal land on cropland in the case of PLUM) subject these to uncertainty analysis. With relatively few internal structural restrictions (e.g. in PLUM the growth rate of yields is not restricted but solely a function of closing the yield gap due to investments in and distribution (global parameters) of in agricultural technology) one would imagine a wider range of outcomes. However, narrower outcome distributions (with a higher or lower central value) could be expected if data becomes available that would allow to reduce uncertain assumptions in exciting parameters greatly or if a new, well-constrained process would be introduced (making the model more complex). The issue of model complexity vs. model performance is addressed in the discussion in the updated text.

b) That PLUM reproduces land use change between 1990-2010 counts in its favor, but cannot really be judged without the specific assumptions made in the validation runs, and does not guarantee much for land use change on longer time scales in a potentially very different environment in which fundamental parameters may change. What needs to be considered here is whether the structure assumed within PLUM is likely to remain unchanged and potential structural changes can explicitly captured within the input parameters. If that is the case, the comparatively reduced structure of PLUM may be an asset that may be exposed more clearly.

Reply: This is a valid point and one that is a challenge for all models and future projections, as it is difficult to identify those processes and mechanisms that will hold from present into the future. One could possibly argue that this challenge might even be larger with increased model complexity. In any case, this is one of the chief reasons why models cannot be used for predictions. For instance, PLUM would fail in producing results that arise due to large structural changes, such as for example the changes in the Russian agricultural system after the breakdown of the Soviet Union (Engström et

al., 2016) – no global land-use model could do that. However, the relative simplicity of PLUM makes it possible to test a wide parameter range consistent with the characteristics of scenarios, and also the uncertainties attached to the scenarios themselves or interpretation of the scenarios. In fact this is one of its chief design principles. This systematic exploration of scenario uncertainties leads to cropland futures that are comparable with the range otherwise typically spanned by several models and scenarios (Alexander et al., under review; Prestele et al., 2016). In view of the reviewer's question, a critical issue is likely the model's simplified accounting for trade, as there is a certain path-dependency for importing and exporting countries. A major shift in trade policies is not easily captured by the model at the moment. However, the increase in production is restricted by the availability of arable land, a variable that of course might change somewhat in the future due to changing climate conditions or new farming technologies, but which in general is rather fixed and gives stability to the trade mechanism. We clarified these aspects in the revised manuscript. All underlying assumptions in PLUM are made transparent in the model description paper (Engström et al., 2016), which helps users to judge the plausibility for future conditions.

My other concern is mainly with the exposition of your research in the manuscript. It lacks a bit of detail on some important assumptions and is unclear on others. My specific remarks in this respect:

1. section 1,2: I think the understandability and readability of your article would benefit from restructuring. You should mention earlier and consistently from the beginning that you are going to compare five future scenarios which correspond to the five SSPs. The probability distributions/ranges assumed for uncertain parameters - including the RCPs - are depending on the scenario and differ between scenarios. You might also want to consider reordering subsections in the Methodology section (2), starting with a discussion of the scenario/uncertainty approach (currently 2.2) and only then give details on the PLUM model (currently 2.1). This would proceed from the more general to the more specific and follow the structuring of the introduction. In section 2.2 (original

numbering), you should then start with something like. "We construct five conditional probabilistic futures (F1-F5), each constructed based on the qualitative and quantitative information in the corresponding SSP (SSP1-5). ... The extent of climatic change within each future was determined by assigning a SSP-specific probability ot each RCP ... " or something similar. I think this would help readers grasp what you are doing at the first reading and right from the start and not having to re-read a few times until that connection becomes clear. There is a clear hierachy between scenario and other input (and RCPs are just one of them) in your scenario construction and your description should reflect that.

Reply: We can see how the suggested re-structuring could improve the readability of our article. We introduced the five future scenarios that correspond with the SSPs in the Introduction. For the method section we followed your suggestion and reordered the subsections accordingly (section 2.2 being now section 2.1).

2. p.2, l.26: I think this sentence is misleading, because basically you use the five SSP scenarios to construct futures. The RCPs are modelled as dependent on SSP, in the same way as many other uncertain input factors. I think it would be helpful for the understanding of the article to make that clear right from the start and introduce the F1-F5 already here.

Reply: We fully agree that the sentence read odd, and will revise this paragraph to: "We developed a scenario-matrix combining the five SSPs with the four Representative Concentration Pathways (RCPs). This scenario-matrix is filled with probabilities based on the assumptions that a given SSP will correspond with a given RCP. For each SSP, we derive RCP-specific input (yields in our case). The resulting conditional probabilistic futures are named F1-F5, where the numbers 1-5 correspond to the SSPs 1-5.".

3. p.3, l.9: A constant cropland-cereal land ratio seems a plausible assumption for 1990-2010, but is there any indication that this will hold for longer time-scales and everywhere around the world?

Reply: We are aware that this approach is not perfect, and include this issue in the discussion. Cropland-cereal land ratios might change with changing demands and changing climate conditions. However, including more crops would considerably increase complexity of the model. It is very challenging to assess the value of increased complexity (the associated level of improvement is very uncertain) vs. the value of having a transparent –and simple- model. We would argue that the country-specific cropland-cereal land ratio is likely to be realistic for the couple of coming decades as the composition of demands or climate conditions are unlikely to change rapidly. For simulations beyond 2050 different demands (e.g. increased demand of lignocellulose crops for second generation biofuels, or changed diets) could change crop composition more substantially (addressed in the revised discussion).

4. p.3, l. 22: It is not clear to me, how actual and potential yields were calculated: You write that the potential yield is the yearly maximum in each grid cell. Does that mean the maximum yield simulated in any year between 2000 and 2100? Or, for each year, the maximum that was obtained for any cell in the country? (Be clear what is per cell and what is per country here and elsewhere.) Likewise, what is the difference to the actual yield? Is the actual yield also simulated or is it observed in the MIRCA dataset directly? The paragraph is not very clear.

Reply: The actual yield is a composite of rain fed and irrigated yields, while potential is the maximum of these for each year and grid cell. This is now clarified in the text. The difference between actual yield and potential yield is obtained by scaling to Mueller (see next comment) and actual yield shows higher variation as it is to a larger extent dependent on rain-fed yields.

5. p.3, l.24 and l. 30: You seem to do a bias correction here, even twice: Once in LPJ-GUESS and once in PLUM, making simulated yields match observed yields. I miss an explanation on why this is necessary and justified, as well as a discussion on which implications this has for the conclusions that can be drawn from your model. (Is it still applicable for extrapolation into the future, i.e. out-of-sample?) (see Ehret et al. 2012,

Hydrology and Earth System Sciences, albeit for a different context)

Reply: The first scaling to Mueller is to obtain actual and potential yield per grid cell. The aggregation to country level, as well as existing differences between the Mueller data-set and FAOSTAT, can cause differences between FAOSTAT and the created actual (country-level) yields. This makes it necessary to scale to FAOSTAT yields, as PLUM is based on FAOSTAT. This is now clarified in the text.

6. p.5, l.31 : Strictly speaking, you are referring to coefficients of variation rather than standard deviations as you are using percentages, aren't you? Or do I misunderstand the paragraph.

Reply: You are correct, now corrected in the text.

7. p.7 l.3; p.7 l.10; p.5, l.17: Although based on established secondary data, there still seems to be quite an "ad hoc" component in the specification of probability distributions for parameters: Is there a justification for assuming normal distributions for all parameters? Is there a justification for assuming independence between input factors. One might argue that uniform or heavy-tailed distributions might better reflect the uncertainty about these distribution themselves and one might also imagine that some parameters could be covariates. E.g. which population projections are assumed by the studies you consulted for projections of global GDP development? Are certain population projections correlated with lower or higher GDP development? Independent sampling from normal distributions for many parameters means that cases where several parameters are at their extremes are unlikely to occur.

Reply: The uncertainty covered by the global parameters can cover several aspects, i.e. the inter-country variability, uncertainty in the development of main drivers in general and uncertainty in the relationships described in PLUM (for example the increase in meat consumption as function of rising incomes). For sake of better knowledge we assumed that the uncertainty of global parameters would be normally distributed around each scenario's central value. This assumption was supported by historic data,

which shows a near-normal distribution in inter-country variability for global parameters (e.g. increase of meat and milk consumption). However, for some scenarios and some parameters we doubted that normal distributions were adequate and chose to truncate the distributions. For the maximum land conversion rates we assumed uniform distributions, as all values seemed equally probable. We have clarified these aspects in the revised main text. We chose to sample the parameters independently, as their degree of correlation in the future are difficult to assess. Further, the use of scenario-specific parameter mean-values deals with the dependencies of parameters, while the parameter ranges address the uncertainties in the exact realisation of the parameter value. The use of the SSP-specific population and GDP development pathways ensures consistency and deals with the correlation of population and GDP development (as the here used GDP development (OECD Env-Growth v9) use the here used population data (IIASA-WiC v9) as input). The imposed uncertainty only covers the range from different models (actually a quite conservative assumption). Additionally, using normal distributions reduces extreme outcomes as the probability of sampling at the tails of several distributions is greatly reduced. Partial correlation is counteracted by the global nature of the parameters, i.e. as it is not very likely that all countries simultaneously encounter the same development, it is more likely that we overestimate the impact of uncertainties rather than underestimate.

8. p.7 l.15 : Did you consider to use an alternative sampling schemes (importance sampling, Sobol' sequences etc.) to increase sampling efficiency?

Reply: No, Sobo'l sequence requires uniform distributions and since PLUM is "cheap" to run thousands of time we did not need to consider more efficient sampling techniques.

9. p.8. l.21ff.: As the description of yield simulations in section 2.1 left me in some doubts, I cannot really follow at this point either. The paragraph here is not clear about your intentions nor the details of implementation in using SVD for analysis and re-sampling. What "pattern resulting from the GCM-RCP yield projections" is to be preserved? Is it the differences between countries? The difference between GCMs? The differences between RCPs? All in all, you do not explain yourself well here...

Reply: The spatial variability introduced by the GCMs is preserved by using the SVD approach. We clarified in the text.

10. p.8. l.24f.: Isn't it SINGULAR value decomposition rather than single value decomposition? Equation 1 is pretty cryptic. Can you use SVD in a formula and add to it? Isn't the outcome of SVD three matrices? Which one is used in this equation here? (Is there any reference for this use of SVD or did you come up with this approach yourselves?)

Reply: Yes, it is singular value decomposition and Equation 1 was wrong (the + sign was not supposed to be there). We excluded the third matrix from Equation 1 as it was replaced with ÉŽ in Equation 2. This is clarified in the text now.

11. p.8. l. 32 (Eq.2): A bit more space between the comma and the second epsilon would greatly increase readability.

Reply: Agreed, changed in Equation 2.

12. p.9. l.14: There is a bit of a confusion about references here. What is the Sobol-jansen method? Is it an R implementation of the method described in Lilburne and Tarantola 2009 or Saltelli 2010 or Engström 2016? Or is it a specific method of GSA? Please clarify and elaborate a bit more.

Reply: The sobol-jansen method is an R implementation of Monte Carlo Estimation of Sobol sensitivity indices, using the method described in Jansen (1999) with improvements of detailed in Saltelli (2010). We clarified this in the manuscript.

13. sections 3.2, 4 and 5: I found your use of convergent and divergent quite confusing. You use it to refer to the fact that three scenarios (F2,F4,F5) have strongly overlapping domains of the outcome distribution, which you describe as convergent, while F1 and F5 are clearly distinct from the each other and from the other three, which you describe as divergent. But, on the other hand, F1 and F3 show rather peaked distributions, which

one might also associate with convergence (uncertainty within the scenario converges around a common mean) and which would be closer (although also not matching) the common use of convergence in statistics, i.e. the convergence to a value as repetitions increase. (F2-F4 show rather wide distributions in contrast that would then rather be divergent. ) Especially, when you speak of a convergent (resp. divergent) scenario, at least I rather have the association of convergence (divergence) WITHIN the scenario and not between several scenarios. At least to me, this caused some confusion, especially since you state "the confidence in the model outcomes for F1 and F3 are the highest, DESPITE the fact that these two scenarios show divergent global cropland development." If you meant convergence/divergence in the statistical sense, this contrast between confidence and divergence would warrant the despite. However, divergence in your use, i.e. the observation that certain scenario assumptions lead to other outcomes than the (slight) majority of other scenario assumptions is not necessarily at odds with higher confidence.

Reply: Thank you for pointing out this issue to us. We understand and have revised the text accordingly.

14. I would avoid the use of convergence/divergence for the pattern you describe. E.g. you could maybe use "strong similarity" instead of "strong convergence" in p.10 l.14, and "represent the extremes of simulated" instead of "show divergent" on p.10,l.10. On p. 11, l.18, you could simply replace "converge" by what you put in parenthesis anyway, and so on.

Reply: Thanks, and we will clarify the text taken these helpful suggestions on board.

15. section 3.3: Why didn't you include the RCPs in Fig.5 (you mention in section 3.1 that their influence is minor, but nevertheless this would complete the picture and put a number to your statement.)

Reply: The GSA tested only the uncertainty in global socio-economic parameters, but not the spatially dependent uncertainty in the input data (yields, in this case). We

clarified this in the text. To quantify the effect of using the mean yield vs. sampling from the matrix, we performed runs with both options. The effect of using the mean yield rather than sampling from the yield matrix was very small (less than 1% in comparison to the more influential variables). So, instead of including a figure that would show the differences of sampling the mean vs. sampling from the yield matrix for global average yield, we included the difference this made for cropland in Figure 3b (dashed vs. solid lines).

16. sections 2,4: While you do an extensive uncertainty analysis on parameters that capture how the future might develop, you seem to use - in several places - model parameters that are rather ad hoc, without deeper justification, e.g. a cereal-to-cropland share of 60%, the assumption that a higher rural share of population means lower production efficiency, etc. These parameters cannot be considered set in stone, especially since you indiscriminately apply them to every country in the world. Why didn't you include a certain variation of these parameters in your uncertainty analysis?

Reply: Cereal-to-cropland share is country specific, but we agree that these ratios can change in the future, see also response to comment 3. It would an interesting exercise to include this parameter in the GSA. The derivation of a plausible uncertainty distribution deserves detailed research of changes in crop distributions and will need to be covered by future research. This limitation is discussed in the revised paper. Regarding the dependency of yields on the share of rural population, this is assumption is part of the yield-gap-closure function, which is based on historic (1990-2010) data. The sensitivity of the coefficient of the relationship in question is tested with the parameter distribution.

17. section 5 (Conclusions): I miss two important aspects in your conclusions: • I miss a short summary of the crucial assumptions incorporated in our model that the obtained results are based on. "Considering current knowledge on crop responses incorporated in crop gorwth model X, a fixed cereal to land share, etc .. , we conclude .. " Especially in uncertainty analysis such as yours, where a lot of parameters are

varied, it is easy to loose track of what are the crucial pillars that the analysis is built on and that give structure to the analysis. You do not vary everything, but there are certain assumptions and model structure that are fixed and bound the results that you obtain. • I also miss a reflection of what we have learned and in how far you were successfull in dissecting the large uncertainty range you metion in the first paragraph of the introduction. Certainly, the uncertainty range was not reduced. So, what have we learned? We know something more about which conditions are necessary to produce the extremes of the distribution. Anything else ...? To what extent is this a new outcome and not yet visible from looking at and disentangling the results of the yet unpublished comparison study (Alexander et al., under review) you cite?

Reply: We agree that our conclusion would be strengthened by including the points raised above and revised the text accordingly.

18. p. 13 l.15f: Again the use of divergent and convergent here is confusing and misleading (see above).

Reply: Revised in the text.

19. Anywhere: What about the Shared Policy Assumptions? Are they reflected anywhere?

Reply: In earlier drafts of the paper we briefly discussed the SPAs and added this in the revised text again.

20. Fig.4: These graphs - as they are - are only readable in color. I could not distinguish different futures on a grayscale print out. If possible, consider placing a scenario label next to each line, then one could also read these graphs in black-and-white.

Reply: Thank you for the advice; we will look into options how to achieve this.

References Alexander, P., Prestele, R., Verburg, P.H., Arneth, A., Fujimori, S., Hasegawa, T., Jain, A.K., Meiyappan, P., Dunford, R., Harrison, P.A., Brown, C., Holzhauer, S., Dendoncker, N., Steinbuks, J., Lenton, T., Powell, T., Sands, R.D., Kyle,

P., Wise, M.A., Doelman, J., Stehfest, E., Schaldach, R., Jacobs-Crisioni, C., Lavalle, C., van Meijl, H., Tabeau, A., Humpenöder, F., Popp, A., Engström, K., Butler, A., Liu, J., Rounsevell, M. (under review) Assessing uncertainties in future land cover projections. Engström, K., Rounsevell, M.D.A., Murray-Rust, D., Hardacre, C., Alexander, P., Cui, X., Palmer, P.I., Arneth, A. (2016) Applying Occam's razor to global agricultural land use change. Environmental Modelling & Software 75, 212-229. Jansen (1999) Analysis of variance designs for model output. Computer Physics Communication 117, 35–43. Prestele, R., Alexander, P., Rounsevell, M., Arneth, A., Calvin, K., Doelman, J., Eitelberg, D., Engström, K., Fujimori, S., Hasegawa, T., Havlik, P., Humpenöder, F., Jain, A.K., Krisztin, T., Kyle, P., Meiyappan, P., Popp, A., Sands, R.D., Schaldach, R., Schüngel, J., Stehfest, E., Tabeau, A., van Meijl, H., van Vliet, J., Verburg, P.H. (2016) Hotspots of uncertainty in land use and land cover change projections: a global scale model comparison. Global Change Biology. Saltelli, A., Annoni, P., Azzini, I., Campolongo, F., Ratto, M., Tarantola, S. (2010) Variance based sensitivity analysis of model output. Design and estimator for the total sensitivity index. Computer Physics Communications 181, 259-270. Schmitz, C., van Meijl, H., Kyle, P., Nelson, G.C., Fujimori, S., Gurgel, A., Havlik, P., Heyhoe, E., d'Croz, D.M., Popp, A., Sands, R., Tabeau, A., van der Mensbrugghe, D., von Lampe, M., Wise, M., Blanc, E., Hasegawa, T., Kavallari, A., Valin, H. (2014) Land-use change trajectories up to 2050: insights from a global agro-economic model comparison. Agricultural Economics 45, 69-84.

---

## Author Comment (AC2) · 25 Jun 2016

We would like to thank both reviewers for their suggestions, and will address these in the revised manuscript. We detail the changes made below.

Response to reviewer 2:

Dear authors, The manuscript provides an interesting insight into the source and magnitude of uncertainty surrounding global crop-land areas under different SSP scenarios. You provide a detailed methodology to quantify uncertainty ranges for various socio-economic parameters.

Reply: Thanks for these supporting comments.

However, I feel the complex methodology is not easy to follow from the way the manuscript is currently structured. My additional comments:

Reply: The method section will be restructured, introducing the modelling framework first. This will enable the reader to get an overview of the entire approach, before we present the more detailed model-set up descriptions and parameter estimates. We clarified the yield description in section 2.2 (previously 2.1) and section 2.4.3, which we think addresses comments 1-4.

1. Does the LPJ-GUESS model used to derive actual and potential yields contain a representation of crop response to heat stress? Under climate scenarios modelled (particularly RCP8.5) temperature changes in several countries will result in certain crops becoming unviable.

Reply: The heat stress implementation is limited to a shortened growing season and increased respiration and lowered photosynthesis, which we clarified in the revised text.

2. Do actual yields from LPJ-GUESS consider the influence of pests and diseases, and how the influence of this may change over time under different climate scenarios?

Reply: The initial actual yields for the year 2000 were derived by scaling LPJ-GUESS actual yields to Mueller et al. Actual yields in Mueller et al. (2012) are observed yields (based on FAOSTAT, national census agencies and agricultural surveys, see (Monfreda et al., 2008)) and are naturally influenced by pests and diseases. As you point out, the influence of pests and diseases might change differently for different climate scenarios, but this was not considered in our study.

3. Following from comments 1 and 2, is the LPJ-GUESS model overestimating actual crop yields (and perhaps potential)? Given that you state agricultural land use is highly sensitive to uncertainties in crop yield growth rates how does this impact your results?

Reply: Initial actual crop yields are not overestimated due to the applied scaling, but

could potentially be overestimated when reported to FAOSTAT. The increase in actual (and potential) yield is derived from the LPJ-GUESS runs and might be optimistic due to the assumed CO2 fertilization.

4. You state scaling factors have been applied to both LPJ-GUESS yields (P3 L24) and yield calculated by PLUM (P3 L30) but it is unlikely that these factors should be constant through time.

Reply: As described in the response earlier, the first scaling establishes differences in yields (actual vs. potential), while the second scaling accounts for potential minor differences in yields from Mueller compared to FAOSTAT. The area of irrigated cropland vs. rain-fed cropland could change over time and result in stronger changes of actual yield. In future work, irrigation scenarios could be included to address this limitation of our study, as well as other management options such as changes in amount and type of fertilisation.

5. The probabilistic futures F1-F5 are not clearly linked to specific SSPs. I also think the manuscript would benefit from a description of each SSP (even if only very brief) within the manuscript in addition to those in the appendix.

Reply: We will introduce the F1-F5 in the introduction and clarify the linkage to the SSPs (p. 2, l. 30). We added a brief description of each SSP.

6. Whilst the study has considered uncertainty of socio-economic variables in great detail, it's unclear how indirect climate uncertainties have been incorporated within the study, for instance, uncertainty within the yield modelling. The study concludes that uncertainties arising from climate variability do not strongly affect the range of global cropland futures but perhaps uncertainty in climate influences are under-represented in the methodology? For the two sets of monte-carlo simulations 3600 runs have been conducted for purely socio-economic parameter investigation (or 720 per SSP) and in conjunction with the 4 RCPs this number increases to 7200 (or 1440 per SSP). So the sampling across an individual SSP-RCP matrix is sparser in comparison.

Reply: We performed one set with 3600 runs using the mean yield of each SSP for each SSP (3600 runs per SSP) and in a second set we did 7200 runs (per SSP), assessing additionally to the socio-economic uncertainties the uncertainties arising from the SSP-RCP matrix and the GCM variability. We clarified this in the revised manuscript.

7. P13 L3: This sentence is confusing. From the previous description of the methodology it seems simulations under RCP8.5 were conducted based on the probability of this scenario within SSPs. Yet this sentence indicates the potential impact of RCP8.5 is under-represented (links to previous comment).

Reply: You are correct; the impact of a high emission pathway is not fully assessed. However, the purpose of this exercise was not to purely assess the impact of each emission pathway on cropland, but to create plausible and consistent cropland futures which address the uncertainties within each scenario. We clarified this in the text.

8. The discussion should explore further why F2, 4 and 5 strongly converge, and why F1 and F3 diverge, linking this to SSP storylines.

Reply: We would like to point out that this was done in section 3.2.

9. Fig 4: It is difficult to distinguish between scenarios in this image, even in colour.

Reply: We added scenario labels next to each line in order to increase clarity.

References Monfreda, C., Ramankutty, N., Foley, J.A. (2008) Farming the planet: 2. Geographic distribution of crop areas, yields, physiological types, and net primary production in the year 2000. Global Biogeochemical Cycles 22. Mueller, N. D., Gerber, J. S., Johnston, M., Ray, D. K., Ramankutty, N., and Foley, J. A. (2012) Closing yield gaps through nutrient and water management, Nature, 490, 254-257.